

# Time-resolved emission reductions for atmospheric chemistry modelling in Europe during the COVID-19 lockdowns

Marc Guevara[1], Oriol Jorba[1], Albert Soret[1], Hervé Petetin[1], Dene Bowdalo[1], Kim Serradell[1], Carles Tena[1], Hugo Denier van der Gon[2], Jeroen Kuenen[2], Vincent-Henri Peuch[3], Carlos Pérez García-Pando[1,4]

[1] Earth Sciences Department, Barcelona Supercomputing Center, Barcelona, 08034, Spain
[2] TNO, Department of Climate, Air and Sustainability, Utrecht, the Netherlands
[3] European Centre for Medium-Range Weather Forecasts, Reading, UK
[4] ICREA, Catalan Institution for Research and Advanced Studies, 08010 Barcelona, Spain

*Correspondence to*: Marc Guevara (marc.guevara@bsc.es)

**Abstract.** We quantify the reductions in primary emissions due to the COVID-19 lockdowns in Europe. Our estimates are provided in the form of a dataset of reduction factors varying per country and day that will allow modelling and identifying the associated impacts upon air quality. The country- and daily-resolved reduction factors are provided for each of the following source categories: energy industry (power plants), manufacturing industry, road traffic and aviation (landing and take-off cycle). We computed the reduction factors based on open access and near-real time measured activity data from a wide range of information sources. We also trained a machine learning model with meteorological data to derive weather-normalised electricity consumption reductions. The time period covered is from 21 February, when the first European localised lockdown was implemented in the region of Lombardy (Italy), until 26 April 2020. This period includes five weeks (23 March until 26 April) with the most severe and relatively unchanged restrictions upon mobility and socio-economic activities across Europe. The computed reduction factors were combined with the Copernicus Atmosphere Monitoring Service's European emission inventory using adjusted emission temporal profiles in order to derive time-resolved emission reductions per country and pollutant sector. During the most severe lockdown period, we estimate the average emission reductions to be -33% for $NO_x$, -8% for NMVOC, -7% for $SO_x$ and -7% for PM2.5 at the EU-30 level (EU-28 plus Norway and Switzerland). For all pollutants more than 85% of the total reduction is attributable to road transport, except $SO_x$. The reductions reached -50% ($NO_x$), -14% (NMVOC), -12% ($SO_x$) and -15% (PM2.5) in countries where the lockdown restrictions were more severe such as Italy, France or Spain. To show the potential for air quality modelling we simulated and evaluated $NO_2$ concentration decreases in rural and urban background regions across Europe (Italy, Spain, France, Germany, United-Kingdom and Sweden). We found the lockdown measures to be responsible for $NO_2$ reductions of up to -58% at urban background locations (Madrid, Spain) and -44% at rural background areas (France), with an average contribution of the traffic sector to total reductions of 86% and 93%, respectively. A clear improvement of the modelled results was found when considering the emission reduction factors, especially in Madrid, Paris and London where the bias is reduced with more than 90%. Future updates will include the extension of the COVID-19 lockdown period covered, the addition of other pollutant sectors potentially affected by the





restrictions (commercial/residential combustion and shipping) and the evaluation of other air quality pollutants such as $O_3$ and $PM_{2.5}$. All the emission reduction factors are provided in the supplementary material.

## 1    Introduction

Since the end of February 2020, most European countries have imposed lockdowns to combat the spread of the COVID-19 pandemic, forcing many industries, businesses and transport networks to either close down or drastically reduce their activity. Such a socioeconomic disruption, which is unprecedented in many ways, has resulted in a sudden drop of atmospheric anthropogenic emissions, including both criteria pollutants and greenhouse gases. The fall of pollutant levels across countries has been identified in multiple studies through the analysis of air quality ground-based and satellite observations (e.g. Bauwens et al., 2020; Collivignarelli et al., 2020; Petetin et al., 2020). While these studies have assessed changes in pollutant concentrations, further understanding of the lockdown impacts upon air quality and climate requires quantifying the reduction of primary emissions. Emissions and weather changes are entangled and looking at concentrations changes only can be largely affected by specific weather conditions, especially considering that the past winter and spring 2020 were exceptionally hot in Europe (C3S, 2020).

Understanding and quantifying the impact of the COVID-19 lockdowns upon European emissions and air quality is difficult due to the heterogeneous implementation of restrictions across different countries, including: (i) different starting dates of the restrictions, (ii) diversity in the levels and type of restrictions, (iii) changes in time of the restriction levels and (iv) different spontaneous response by individuals (e.g. voluntary decision to change the way of commuting). The chronology of the lockdowns is illustrated in Fig. 1, which shows stringency index trends computed by the Oxford COVID-19 Government Response Tracker (OxCGRT) for selected countries (Hale et al., 2020). The stringency index reports how the response of governments varied over several indicators (e.g. school closures, restrictions in movement, implementation of economic policies), becoming stronger or weaker over the course of the COVID-19 pandemic. The analysis of the stringency index trends is focussed on 6 European countries with different lockdown patterns for illustration (Italy, Spain, France, Germany, the United Kingdom and Sweden). As observed, Italy was the country where restrictions first started, followed by Spain and France, where national lockdowns were imposed on 14 and 17 March, respectively. In contrast to Italy, where the transition from low to high stringency levels was gradual, these two countries abruptly experienced severe restrictions on movements, and commercial and industrial activities. A similar pattern is observed for Germany and the United-Kingdom (UK), where national lockdowns were imposed on the 20 and 23 March, respectively. Sweden, on the other hand, was one of the few European countries where no national lockdowns were implemented and only national recommendations (e.g. relatively soft social distancing measures) were provided to citizens. This is clearly illustrated in the evolution of its stringency index, which remained lower than in the other countries during the whole period.



Considering all of the above, the quantification of emission changes due to the COVID-19 lockdown requires the use of reduction factors that are, at least: (i) country-dependent, (ii) pollutant sector-dependent and (iii) daily dependent for some sectors. Some studies focussing on the quantification of emission reductions are beginning to be published. Le Quéré et al. (2020) quantified the reduction in daily $CO_2$ emissions during the COVID-19 lockdown from January 2020 to April 2020 over 69 countries, 50 US states and 30 Chinese provinces for a total of six sectors of the economy (i.e. energy industry,

manufacturing industry, road transport, residential sector, public sector and aviation). The study, which calculates the emission reductions based on national activity data, was focussed on estimating the expected impact of the lockdowns upon the 2020 annual $CO_2$ emissions and climate, but it did not include an analysis of emission cuts of criteria pollutants ($NO_x$, $SO_x$, NMVOC, $NH_3$, PM10 and PM2.5) or air pollution levels. More recently, Menut et al. (2020) developed an emission scenario for Western Europe to quantify the impact of the lockdowns on air quality levels. Although focussing on criteria pollutants, the emission

scenario was limited to March 2020 and was set up using only the Apple movement trends, which were used to derived emission reductions not only for road transport but also for other anthropogenic sources (i.e. manufacturing industry, non-road transport and residential/commercial combustion).

We present an open-source dataset of daily, sector- and country-dependent emission reduction factors for Europe associated

with the COVID-19 lockdowns. These factors are designed to both support the quantification of European primary emission reductions and the associated impacts upon air quality. Our emission reduction factors are based on a bottom-up approach that considers a wide range of information sources, including open access and near-real time measured activity data, proxy indicators and other available reports. The resulting dataset covers from the 21st February 2020, the beginning of localised lockdown in Italy (region of Lombardy), to the 26th April 2020 and the following anthropogenic source categories: energy

industry, manufacturing industry, road transport and aviation (landing and take-off cycle, LTO).

To assure easy adoption of the emission reduction factors they are produced in a format consistent with the CAMS-REG-AP emission inventory developed under the Copernicus Global and Regional emissions service (CAMS_81) (Kuenen et al., 2014; Granier et al., 2019), whose main objective is to provide gridded distributions of global and European emissions in direct

support of the Copernicus Atmosphere Monitoring Service (CAMS) production chains (Marécal et al., 2015; Huijnen et al., 2019; Rémy et al., 2019). In the framework of CAMS, the CAMS-REG-AP emission inventory is currently used by several modelling services, mainly to provide short term air quality forecasts, long-term air quality re-analysis or policy support products. To illustrate the potential application of our reduction factors, we also performed air quality simulations to quantify and evaluate the observed changes in $NO_2$ concentrations across Europe. We considered three emission scenarios: (i) a first

one with business as usual emissions using the default CAMS-REG-AP inventory, (ii) a second one considering only the traffic-related emission reductions, and (iii) a third one including the reductions from all the aforementioned sectors. The difference between scenarios allows quantifying the impact of the lockdown measures on emissions and air quality levels and, particularly, the contribution of the road transport activity to the overall reductions. The study period of these modelling





exercises covers one month prior to the first day of lockdown in Italy (20 January to 20 February) and more than two months

of COVID-19 lockdown conditions (21 February to 26 April). Therefore, the focus of the work is on the transition to full lockdown conditions. The process toward normal conditions is a still ongoing process and will be assessed in future works.

Section 2 describes the methods and datasets used to estimate the European emission reduction factors for each one of the aforementioned pollutant sectors. Section 3 describes the setup of the modelling experiment to test the performance of the

reduction factors on modelling the decrease of emissions and $NO_2$ concentrations across Europe. Section 4 discusses the results obtained in terms of emissions and $NO_2$ level reductions. Section 5 includes our main conclusions and perspectives for future updates.

## 2    Time-, country- and sector-resolved emission reduction factors

We computed a set of emission reduction factors for Europe that vary per day, country and sector. The resulting dataset follows

the sector classification reported by the CAMS-REG_AP emission inventory, which corresponds to the Gridded aggregated Nomenclature For Reporting (GNFR). We considered four GNFR sectors, GNFR_A (energy industry), GNFR_B (manufacturing industry), GNFR_F (road transport) and GNFR_ H (aviation), which we assumed to be the ones suffering the largest reduction in their activity during the COVID-19 lockdowns, in line with Le Quéré et al. (2020). Other sectors potentially affected by the COVID-19 lockdown such as GNFR_C (other stationary combustion activities) or GNFR_G (shipping) were

not included in this first assessment and will be addressed in future releases of the dataset.

In terms of spatial coverage, we included as many countries as possible that are covered by the CAMS-REG_AP European working domain (30° W – 60° E and 30° N – 72°N) (a complete list of the countries can be found in Granier et al. (2019)), giving a special priority to EU-30 (EU-28 plus Norway and Switzerland). A list of the countries included for each sector is

summarised in Table 2. The time span of the reduction factors is from 21 February to 26 April 2020. The beginning of the period corresponds to the date of the first localised lockdown in the region of Lombardy, Italy. Three distinct phases can be identified from the OxCGRT stringency index trends in Fig. 1: (i) a first phase without restrictions, with the exception of Italy (1st January to 12th March), (ii) a second phase with increasingly severe restrictions (12 to 23 March) and (iii) a third and final phase when the restrictions were at their maximum and remained almost unchanged for five weeks (23 March to 26 April).


We collected and processed daily measured time-series representing the main activities of each sector. We then combined this information with specific methods in order to derive daily emission reduction factors as a function of the country and sector. Table 1 summarises the main sources of information used and the countries included for each sector. The following subsections describe the data and methods for each sector along with the underlying assumptions.


## 2.1 Energy industry

We assumed the changes in emissions from the energy industry (which includes power and heat plants) to follow the changes observed in the electricity demand data reported by the European Network of Transmission System Operators for Electricity (ENTSO-E) transparency platform (Hirth et al., 2018; ENTSO-E, 2020). ENTSO-E centralizes the collection and publication of the electricity generation for each European Member State. For each country, we collected daily electricity demand data for years 2015 to 2020 (January to April). Data gaps and inconsistencies found in the original dataset were corrected using the electricity generation statistics reported by the national Transmission System Operators (TSOs). For Russia, we derived the electricity demand data directly from Russia's Federal Grid Company of Unified Energy System (FGC UES, 2020).

In addition to its characteristic weekly variability, with higher values during weekdays, part of the electricity demand is driven by temperature fluctuations. Therefore, to calculate the reduction in electricity demand during the COVID-19 lockdowns, we first estimated the business-as-usual (BAU) electricity demand, i.e., the demand that would have occurred in the absence of lockdowns under the same meteorological conditions. To estimate the BAU electricity demand we used ML models trained with meteorological data and other time features. This approach has been used to weather-normalize $NO_2$ surface concentration time-series, whose variability is also partly driven by the meteorological conditions, to quantify actual reductions of $NO_2$ during the COVID-19 lockdown (Petetin et al., 2020). More specifically, we used gradient boosting machine (GBM) models trained and tuned independently for each country using daily data from January to April between 2015 and 2019. As inputs, we considered the following features: country-level daily population-weighted Heating Degree Days, date index (number of days since 2015/01/01), Julian date, day of week and a Boolean feature indicating the country-specific bank holidays. The HDD is defined relative to a threshold temperature ($T_b$) above which a building needs no heating and is used to approximate the daily energy demand for heating a building (Quayle and Diaz, 1980). In order to provide a more realistic estimate of the potential electricity demand for space heating on a national level, we computed country-specific population-weighted HDD values ($HDD\_pop(d)$) following Eq. (1):

$$HDD\_pop(d) = \sum_{x=1}^{n} \frac{(\max{(T_b - T_{2m}(x,d),0)}) * Pop(x)}{\sum_{x=1}^{n} Pop(x)} \qquad (1)$$

Where $T_{2m}(x,d)$ is the daily mean 2 meter outdoor temperature for grid cell $x$ and day $d$ [°C]; $Pop(x)$ is the amount of population included in grid cell $x$ [n° of inhabitants] and $n$ is the total number of grid cells that corresponds to a specific country. A threshold temperature value of 15.5°C was selected following Spinoni et al. (2015). Outdoor temperature information was obtained from the ERA5 reanalysis dataset for the period 2015 – 2020 (C3S, 2017), while information on gridded population was derived from the Gridded Population of the World, Version 4 (GPWv4; CIESIN, 2016). Each grid cell was assigned to a specific country following the global country mask available in the Emissions of atmospheric Compounds and Compilation of Ancillary Data system (ECCAD, https://eccad.aeris-data.fr/).





Julian day and day of week serve here as proxies for the (climatological) main drivers of the seasonal and weekly variability

of the power demand, and the date index acts as the trend term. We replicated the tuning strategy previously used in Petetin et al. (2020) with random search in the hyper-parameter space and rolling-origin cross-validation (appropriate for time series). While the training and tuning of the GBM models was performed from 2015 to 2019, we used the two first months of 2020 (January-February) to test the performance of the models.

Figure 2 summarizes the main statistics (normalized mean bias, NMB; normalized root mean square error, NRMSE and correlation, r) obtained from the comparison between measured and ML-based electricity demand during the first two months of 2020 for selected countries. Generally, a high correlation (above 0.9) and low NMB and NRMSE (below 5%) are observed for all cases, especially in those countries with stronger lockdown restrictions such as Italy, France or Spain. The poorest performance was obtained in Finland (r = 0.33), due to a strong negative anomaly (-12% on average) of electricity demand in

January-February 2020 compared to previous years used for training. Compared to most other countries, a larger NRMSE and lower correlation was also found in Luxembourg. In addition, despite relatively good statistics in early 2020, the electricity demand computed in Denmark and Norway shows a substantial and unexpected increase during the COVID-19 lockdown (up to +12%). As a precautionary measure, we assumed a null reduction of the electricity demand in Denmark, Finland and Norway, and a fixed -16% reduction in Luxembourg starting the first day of the national lockdown implementation (15th of

March), following the results reported by Le Quéré et al. (2020).

The electricity demand started to decrease by the end of February and the beginning of March 2020 compared to the BAU electricity demand estimated from the GBM models in countries where strong restrictions had been implemented. We attributed these discrepancies to the direct effect of lockdown measures, regardless of the meteorological conditions, and used

them to derive quantitative daily emission reduction factors for the energy industry sector (Eq. 2)

$$RF_{ener\_indu}(d,c) = \left( \frac{ED_{COVID-19}(d,c) - ED_{measured}(d,c)}{ED_{measured}(d,c)} \right) * 100 \qquad (2)$$

where $RF_{ener\_indu}(d,c)$ is the final reduction factor for the energy industry sector for day $d$ and country $c$ [%];

$ED_{COVID-19}(d,c)$ is the estimated BAU electricity demand computed using ML for day $d$ and country $c$ [MW] and $ED_{measured}(d,c)$ is the measured electricity demand for day $d$ and country $c$ [MW].

Figure 3.a illustrates the reduction factor trends obtained for selected countries. As expected, the strong weekly cycle of electricity demand normally observed in most countries smoothed down during the COVID-19 lockdown. The resulting trends

are consistent with the national lockdown calendars and restriction levels implemented in each country. Italy is the first country



where traffic activity reductions happened, followed by Spain, France, Germany, UK and Sweden. This is in line with the starting dates of lockdown restrictions in each country (Sect. 2). For Spain, reduction increased between March 30th and April 9th, the most restrictive phase of the Spanish lockdown when only essential activities including food trade, pharmacy, and some industries were authorized. In the case of Sweden, positive values are observed for certain days until the beginning of April.

These results agree with the ones reported in Le Quéré et al. (2020), who obtained a 4% increase during the lockdown for this country. It is likely that electricity demand from public and commercial services remained unperturbed as, in contrast to most countries, there was no enforced lockdown in Sweden. We also hypothesize that a voluntary self-isolation of a fraction of the population may have increased household electricity consumption. During the strictest period of the COVID-19 lockdown (23 March – 26 April), Italy was the country experiencing the largest reductions (-21%), followed by Spain (-15%) and France (-

205 14.4%).

The countries for which daily reduction factors could be computed are shown in Table 1. For countries with no data, we constructed a set of reduction factors based on the average data of all the available countries except Italy, where the lockdown restrictions began approximately 3+ weeks before other countries.

**2.2    Manufacturing industry**

The reduction factors for manufacturing industry are based on the daily electricity demand reduction factors described in Sect. 2.1. We attributed 25% of the total electricity demand reduction to the reduction in manufacturing industry activity. We estimated this value considering that: (i) the European industry sector consumes 22.3% of the total final electricity demand (Eurostat, 2020a) and (ii) most of the electricity reduction during the lockdown can be linked to commercial and public

services. Indeed, the manufacturing industry sector has maintained certain activities during the COVID-19 pandemic, in contrast to the commercial and public services sectors that were forced to reduce or even completely halt their activities (e.g. restaurants and hotels, office buildings). Figure 4 shows, on the one hand, the evolution of the Industrial Production Index (IPI) for selected industrial branches in Spain between January 2019 and April 2020 (INE, 2020) and, on the other hand, the contribution of each Spanish commercial and public service branch to the total electricity consumption (IDAE, 2018). While

certain industrial branches have suffered important decreases on their production levels during March and April 2020 (i.e. production of mineral products, steel industry), the essential ones kept about the same level of productivity (i.e. pharmaceutical preparations, manufacturing of soap and detergents, petroleum refining). In contrast, office and commercial buildings, schools, universities, restaurants and hotels, which represent more than 70% of the total electricity consumption, were obliged, in most cases, to close their facilities during the lockdown.


The reduction of power demand attributable to the manufacturing industry sector was then translated into a total reduction in industrial activity using the national energy balances reported in Eurostat (2020a) (Eq. 3):



$$RF_{manuf\_indu}(d,c) = \frac{RF_{ene_{indu}}(d,c)*0.25}{S_{indu}(c)} \tag{3}$$


where $RF_{manuf\_indu}(d,c)$ is the final reduction factor for the manufacturing industry sector for day $d$ and country $c$ [%], $RF_{ene_{indu}}(d,c)$ is the reduction factor for the total electricity demand for day $d$ and country $c$ estimated as described in Sect. 2.1 [%], and $S_{indu}(c)$ is the share of final electricity consumed by the industrial sector in country $c$ [%] (Eurostat, 2020a).

Figure 3.b shows the daily reduction factors computed for selected countries. The original positive values (i.e. increase of electricity consumption) obtained for the energy industry sector (Fig. 3.b) were replaced by zeros for the calculations, as we consider unlikely that average increases in manufacturing industrial emissions occurred during the lockdown. In general, the trends observed in all countries follow the same pattern as the ones presented for the energy industry. During the strictest period of the COVID-19 lockdown, computed reductions are between -13 and -10% for Italy, Spain, France and UK, -4% for
Germany and -0.8% for Sweden.

## 2.3  Road transport

The emission reduction factors considered for the road transport sector are based on the Google COVID-19 Community Mobility Reports (Google LLC, 2020). The Google dataset reports daily movement trends over time by geography (country and region) across different categories of places (i.e. groceries and pharmacies, parks, transit stations, retail and recreation,
residential and workplaces) based on aggregated and anonymized sets of data from users who have turned on the Location History setting for their Google Account on their mobile devices. For the present study, we used the mobility trends reported for the transit stations category, which includes places like public transport hubs such as subway, bus, and train stations. The assumption behind this choice is that movement trends observed in public transport hot-spots correlate with private transport trends. Reductions for each day are calculated by Google from a baseline taken as the median value, for the corresponding day
of the week, over a 5-week period prior to the lockdowns (3 January to 6 February).

We evaluated the Google movement trends with actual measured traffic counts from the city of Barcelona (ATM, personal communication) and other major interurban roads in Spain (DGT, 2020), the latter discriminated by vehicle type (light- and heavy-duty) (Fig. 5). Note that for the Barcelona and DGT data, the information is available from 3 and 9 March onwards,
respectively. In general terms, Google data reproduce the measured-based trends obtained for the city of Barcelona (BCN) and the Spanish interurban roads (DGT-all), with correlations of 0.96 and 0.92, respectively. Overall, the average reductions reported by each of these three datasets are similar: -74.6% (Google), -69.1% (BCN) and -63.62% (DGT-all). Using Google data at transit stations tends to slightly overestimate the reductions observed during the weekdays. However large discrepancies are shown when comparing the Google trend against the one reported by DGT for heavy-duty vehicles (DGT-heavy). The data
from the DGT reports an average reduction of heavy-duty vehicles of only -31% (more than 2 times lower than the one reported





by Google), as these vehicles supported the delivery of essential goods and products (e.g. food, medical supplies). Nevertheless, we omitted the distinction between light and heavy-duty vehicle when developing the reduction factors because CAMS-REG_AP/GHG traffic-related emissions are not discriminated by type of vehicle. Consequently, our factors for the traffic sector may overestimate the overall reduction of emissions, especially in areas with a higher share of heavy-duty vehicles,

typically interurban roads, and for pollutants such as PM that are emitted in a higher proportion from those vehicle categories. This approach may be improved in the future but was constrained in this study by data availability.

Figure 3.c shows the reduction factors proposed for selected countries. As in the case of energy industry, the resulting trends are in line with the implementation and evolution of the national restrictions imposed in each country. The decrease of the

traffic activity in Italy starts two days after the implementation of the localized lockdown and intensified once the national lockdown was imposed on 12 March, reaching reductions of about -80%. In the case of Spain and France, similar traffic reduction levels were reached just 3 days after the beginning of the corresponding national lockdowns. For UK and Germany, the largest reductions are around -70% and -50%, respectively. The lower reductions in Sweden (around -40%) are consistent with the lack of enforced mobility restrictions in this country at any point. In all cases, the activity started recovering during

the last week of the period of study, coinciding with the relaxation of the mobility restrictions.

The list of countries included for this sector is summarised in Table 1. For countries without available data we constructed a set of average reduction factors considering all countries except Italy.

### 2.4    Aviation

We derived the reduction factors related to air traffic emissions during Landing and Take-Off cycles (LTO) in airports from statistics provided by FlightRadar24 (FlighRadar24, 2020), which reports every day the total number of tracked operations per airport over the preceding 30 days. For each country, we selected the largest airport to represent a national proxy. We computed country specific daily flight operation trends using as a baseline value the average number of operations per airport from the previous year reported by Eurostat statistics (Eurostat, 2020b).


We started collecting the information from FlightRadar24 for all airports on 6 March, and the information from previous dates could not be retrieved as it is not archived. Therefore, our reduction factors have as initial date the 6 March in all cases, independently of the lockdown calendars. As shown in Fig. 3.d for most countries the reductions in flight activity were starting to occur during those dates and therefore the trends presented are consistent. However, in some other countries such as Italy,

reductions were already in a more advanced state (first day of reduction is -15%). We do not expect this lack of information to affect significantly the emission and air quality modelling results, as the contribution of this pollutant sector to total European emissions is very low, i.e. 1.1% and 0.14% to total $NO_x$ and PM10 emissions, according to the last available EMEP official reported emission data (EMEP/CEIP, 2020). We expect to complement this information from alternative sources of data in a





future release of the dataset. Regarding the obtained results, it is observed that in almost all countries, the reduction levels
reached values of -90% or more before the beginning of April. In contrast to road transport, there were no signs of recovery
during the last week of April for this sector, as the movements between countries were still restricted at that time.

## 3 Evaluating the reduction factors with air quality modelling

We performed an emission and air quality modelling study as a first demonstration and evaluation of the applicability of the
developed emission reduction factors. We used the Multiscale Online Nonhydrostatic AtmospheRe CHemistry model
(MONARCH) (see section 3.1) and the High-Elective Resolution Modelling Emission System version 3 (HERMESv3) (Sect.
3.2) both developed at the Barcelona Supercomputing Center. The simulation period for the case study is from 20 January to
26 April 2020. The study period covers one month of pre-COVID lockdown conditions (the first localised lockdowns in Europe
began on 21 February in the region of Lombardy) and more than two months of lockdown conditions, including five weeks
(23 March to 26 April) during which the most severe restrictions were already implemented in most (22) European countries.
Therefore, the selected period of study allows analysing the changes in concentrations between the lockdown period and before.

Three air quality simulations were run: (i) using the default CAMS-REG-APv3.1 emissions without considering any emission
reduction, hereafter referred to as *baseline* scenario, (ii) considering the traffic-related emission reduction factors only,
hereafter referred to as *covid19_traffic* scenario, and (iii) including the reduction factors from the traffic, energy and
manufacturing industry and aviation sectors, hereafter referred to as *covid19_all* scenario. We also compared the model results
against measurements of the European Environmental Agency (EEA) AQ e-Reporting (EEA, 2020) available through the
Globally Harmonised Observational Surface Treatment (GHOST) project (Sect. 3.3). The model and evaluation work focuses
on $NO_2$. Given that our main focus are the emission reductions and their evaluation, the inclusion of other relevant, yet more
model-dependent secondary pollutants such as $O_3$ or $PM_{2.5}$ is beyond the scope of this paper. The impact of the lockdown upon
secondary pollutants, which are affected by more complex chemical interactions and source contributions, may be addressed
in a follow-up multi-model study.

### 3.1 MONARCH model

MONARCH v1.0 (Pérez et al., 2011; Haustein et al. 2012; Jorba et al., 2012; Spada et al., 2013; Badia and Jorba, 2015; Badia
et al., 2017) is a fully online integrated system for meso- to global-scale applications developed at the Barcelona
Supercomputing Center (BSC). A flexible gas-phase module combined with a hybrid sectional-bulk multicomponent mass-
based aerosol module is implemented in the MONARCH model, that uses the Nonhydrostatic Multiscale Model on the B-grid
(NMMB; Janjic and Gall, 2012) as the meteorological core driver. The Carbon Bond 2005 chemical mechanism (CB05;
Yarwood, 2005) extended with Toluene and Chlorine chemistry is the gas-phase scheme used in MONARCH. The CB05 is
well formulated for urban to remote tropospheric conditions and it considers 51 chemical species and solves 156 reactions.



The rate constants were updated based on evaluations from Atkinson et al. (2004) and Sander et al. (2006). The photolysis scheme used is the Fast-J scheme (Wild et al. 2000). It is coupled with physics of each model layer (e.g., aerosols, clouds, absorbers as ozone) and it considers grid-scale clouds from the atmospheric driver. The Fast-J scheme has been updated with CB05 photolytic reactions. The quantum yields and cross section for the CB05 photolysis reactions have been revised and updated following the recommendations of Atkinson et al. (2004) and Sander et al. (2006). The aerosol module in MONARCH

describes the lifecycle of dust, sea-salt, black carbon, organic matter (both primary and secondary), sulfate and nitrate aerosols. While a sectional approach is used for dust and sea-salt, a bulk description of the other aerosol species is adopted. A simplified gas-aqueous-aerosol mechanism has been introduced in the module to account for the sulfur chemistry, the production of secondary nitrate - ammonium aerosol is solved using the thermodynamic equilibrium model EQSAM, and a two-product scheme is used for the formation of secondary organic aerosols from biogenic gas-phase precursors. Meteorology driven

emissions are computed within MONARCH. Mineral dust emissions are calculated with an updated version of Pérez et al. (2011) scheme, the sea salt aerosol emissions following Jaeglé et al. (2011), and biogenic gas-phase species using the MEGANv2.04 model (Guenther et al., 2006). The model provides operational regional mineral dust forecasts for the World Meteorological Organization (WMO; https://dust.aemet.es/), and participates to the WMO Sand and Dust Storm Warning Advisory and Assessment System for Northern Africa-Middle East-Europe (http://sds-was.aemet.es/). Since 2012, the system

contributes with global aerosol forecast to the multi model ensemble of ICAP initiative (Xian et al., 2019) and since 2019, it is a candidate model of the CAMS - Air Quality Regional Production (Marecal et al., 2015).

In this work, the model is configured for a regional domain covering Europe and part of northern Africa. The rotated lat-lon projection is used, with a regular horizontal grid spacing of 0.2 degrees, and the top of the atmosphere is set at 50 hPa using

vertical layers. Figure S1 displays the domain of study. Meteorological initial and boundary conditions were obtained from the ECMWF global model forecasts at 0.125 degrees and chemical boundary conditions from the CAMS global model forecasts at 0.4 degrees (Flemming et al., 2015). For an efficient execution of the modelling chain, the autosubmit workflow manager is used (Manubens-Gil et al., 2016).

### 3.2 HERMESv3 emission system

The original annual CAMS-REG-APv3.1 emission inventory was processed using the HERMESv3 system, an open source, stand-alone multi-scale atmospheric emission modelling framework developed at the BSC that computes gaseous and aerosol emissions for use in atmospheric chemistry models (Guevara et al., 2019). The HERMESv3 system was used to remap the original CAMS-REG-AP data (0.1x0.05 degrees) onto the MONARCH modelling domain and to derive hourly and speciated emissions. Aggregated annual emissions were broken down into hourly resolution using the emission temporal profiles

reported by Denier van der Gon et al. (2011). The speciation of NMVOC and PM emissions was performed using the split factors reported by TNO (Kuenen et al., 2014).





For the *covid19_traffic* and *covid19_all* scenarios, the estimated reduction factors (Fig. 3.a,b,c,d) were combined with the original temporal profiles in order to model dynamic emission reductions for each sector and country. For each pollutant sector,

we constructed a dataset of country-specific COVID-19 daily temporal profiles by combining the original temporal weight factors reported by Denier van der Gon et al. (2011) with the computed emission reduction factors, following Eq. (4):

$$DF\_covid19_s(c,d) = DF_s(d) * \left(1 + \frac{RF_s(c,d)}{100}\right) \tag{4}$$

where $DF_s(d)$ are the daily temporal factors for pollutant source $s$ and day of the year $d$ [0 to 366], and $RF_s(c,d)$ is the reduction factor computed for sector $s$, day of the year $d$ and country $c$ [%]. The $DF_s(d)$ weight factors were obtained by combining the original monthly (January to December) and weekly (Monday to Sunday) temporal profiles reported by Denier van der Gon et al. (2011). Figure 3.e illustrates the COVID-19 daily temporal factors for the road transport sector in selected countries. The original daily profile for this sector, which is used in *baseline* scenario, is also plotted for comparison purposes.

In general, the temporal disaggregation of emissions would require the sum of the daily weight factors to be 366 (as in this case the year of study is a leap year). Nevertheless, and due to the application of the reduction factors, the sum of the COVID-19 daily factors do not add up to this number, which allows simulating time-resolved emission reductions.

### 3.3    Observational dataset

The GHOST project is a BSC initiative dedicated to the harmonisation of publicly available global surface observations (most

notably air quality pollutants) and metadata, for the purpose of facilitating a greater quality of observational/model comparison in the atmospheric chemistry community (Bowdalo, in preparation). Numerous networks are currently processed and contained under the umbrella of GHOST including, among other, the EBAS and EEA networks. For each network, all relevant numerical and textual metadata (e.g. station classifications, measurement methodologies) is standardised and all data is passed through numerous quality control tests, giving detailed quality assurance (QA) flags.


In this work, we used the $NO_2$ near-real time EEA data. We selected rural and urban background stations located at selected countries (Italy, Spain, France, Germany, UK and Sweden). In the case of urban background stations, we selected those located in Milano, Madrid, Paris, Berlin and London. For Sweden, and due to the low density of stations found in individual cities (e.g. Stockholm, 1 station), we decided to consider all urban background stations available country wise (6). GHOST provides

a wide range of harmonized metadata and quality assurance (QA) flags for all pollutant measurements. In this study, we took benefit of these flags to apply an exhaustive QA screening. More details on the QA flags used can be found in Appendix A. Note that for Italy, there is a data gap between 1 February and 13 February in all stations. We nevertheless decided to keep this country in our evaluation study since it is one of the European countries most affected by the COVID-19 pandemic and the data gap does not affect the lockdown period. In the case of Sweden, only 1 rural background station was available for the





entire country, which may reduce the representativity of the computed results. A detailed description of the stations is available
in Table S1 and Fig. S1 of the supplementary material.

## 4 Results and discussion

Figure 6 shows maps of daily average $NO_x$ emissions [kg·s$^{-1}$·m$^{-2}$] and $NO_2$ concentrations [µg·m$^{-3}$] obtained for the *baseline*
scenario between 23 March and 26 April, as well as the differences with respect to the *covid19_all* scenario (i.e. *covid19_all*
*minus baseline*). During this 5-week period most European countries were under severe national lockdown restrictions, which
allows illustrating the largest impacts upon emissions and air quality levels.

For both $NO_x$ emissions and $NO_2$ concentrations, the main reductions occurred in urban areas and main interurban roads,
especially within the most affected countries (i.e. Italy, Spain, France, the UK). The largest emission reductions are related to
traffic (Sect. 4.1), which is the main contributor to urban $NO_2$ levels, with approximately a 40% share on average (EEA, 2019).
Below we discuss the results obtained from the modelling experiments in terms of daily changes in emissions (Sect. 4.1) and
$NO_2$ air quality concentrations (Sect. 4.2) during the study period.

### 4.1 Emissions

Figure 7 (a, b, d, c) shows the evolution of daily $NO_x$, NMVOC, $SO_x$ and $PM_{2.5}$ emissions during the entire period of study (20
January to 26 April) for EU-30 and for each of the three scenarios. The largest emission reductions occurred during the second
and third week of March, when several European countries enforced national lockdown restrictions. After this period, there
was a stabilization of the emission reductions until approximately the 19 April. Thereafter, a slight recovery of the emission
levels started to occur, which is consistent with the recovery of traffic activity shown in Fig. 3.c. Overall, and when comparing
the *baseline* and *covid19_all* scenarios, the reduction of total emissions is -33% for $NO_x$, -8% for NMVOC, -7% for $SO_x$ and
-7% for PM2.5. The contribution of the traffic sector to total reductions is especially relevant for $NO_x$ (90%), NMVOC (87%)
and PM2.5 (82%) while for $SO_x$ most of the total reduction can be attributable to the decreases in the energy and manufacturing
industries (97%), according to the results shown by the *covid19_traffic* scenario. Figure 7 (e, f) illustrates the average and
5th/95$^{th}$ percentiles (p05, p95) of the daily relative changes [%] in the gridded $NO_x$ emissions for Italy and Sweden. The results
were computed considering all the grid cells within each of the countries. In Italy, the last two weeks of March and first two
weeks of April certain shows areas of the country reaching reductions up to -75%, whereas in other areas less affected by
anthropogenic (and particularly road transport) emissions the reductions were significantly lower (~ -25%). In the case of
Sweden, the reductions ranged between -6% (p95) and -36% (p05).

Figure 8 summarises the average, minimum and maximum national daily emission changes [%] obtained for $NO_x$, NMVOC,
$SO_x$ and PM2.5 between 23 March and 26 April for selected countries along with the average at EU-30 level. Changes in





emissions present strong variations from country to country and pollutant to pollutant. For $NO_x$ and $SO_x$, all countries except Germany and Sweden present stronger average reductions than the ones reported at the EU-30 level (-33% and -7%, respectively), and Italy and France are the two countries with the largest reductions (-50% for $NO_x$ and -12% for $SO_x$). For $NO_x$, minimum and maximum daily emission reductions are in general relatively close to the average (e.g. Italy: avg = -50%,

min = -47% and max = -56%; Spain: avg = -40%, min = -43% and max = -46%). In contrast, there are large differences among the average, minimum and maximum daily $SO_x$ changes, especially in Germany (Sweden) where changes in emissions go from 0.6% (0.15%) to -12% (-5%). The different behaviours observed for $NO_x$ and $SO_x$ are related to the different trends of the road transport and energy industry (Fig. 3.a and c). The daily variability of the reduction factors for road transport is generally low; in the case of the energy industry large day-to-day variations are observed.


Despite having experienced one of the largest reductions in road transport activity (more than -80%), Spain was the country with the lowest decrease in total PM2.5 emissions (-4.3%), and the second lowest in terms of NMVOC (-4.4%). Sweden shows a PM2.5 emission reduction of -7.6%, almost two times larger than Spain and very close to Italy (-9.2%), despite its lower traffic activity decrease (less than -40%). This is explained by the different contributions of the road transport sector

contribution to total emissions in each country. Figure 9 shows the relationship between the reduction of traffic activity and contribution of the road transport sector to total emissions per country and pollutant. In the case of Sweden, road transport represents around 21.3% of total PM2.5 emissions, while in Spain the contribution is just 7.9%. Similarly, in the case of NMVOC emissions the contribution of road transport emissions is 15.9% in Italy and 8.9% in France, while in Spain is only 4.3%.


## 4.2    Air quality

Figure 10 shows the observed and modelled hourly $NO_2$ concentrations between 20 January and 26 April at selected urban background sites in Italy (Milano), Spain (Madrid), France (Paris), UK (London), Germany (Dusseldorf) and Sweden (all available sites). In the same way, the results at rural background stations are presented in Fig. 11. In both cases, the results are

presented separately for each of the emission scenarios considered: *baseline* (in magenta), *covid19_traffic* (in green) and *covid19_all* (in blue). Statistical parameters computed on an hourly basis (i.e. mean bias, MB; root mean square error, RMSE; correlation coefficient, r) are presented for each emission scenario, country and station type for the pre-lockdown (20 January to 20 February) and most restrictive lockdown period (23 March to 26 April) (Fig. 12). (For the pre-lockdown period, the calculated statistics are equal for all scenarios, as no emission reductions are considered during that time.) The computation of

statistics during the pre-lockdown period allows quantifying the performance of the system under BAU conditions. Table 2 summarises the absolute and relative changes of $NO_2$ concentrations at each station type and country between 23 March and 26 April.



The MONARCH model is capable of reproducing fairly well the urban background $NO_2$ observations during the pre-lockdown

period, particularly in London (MB = -0.25 µg·m⁻³, RMSE = 16 µg·m⁻³, r = 0.74), Madrid (MB = -4 µg·m⁻³, RMSE = 19 µg·m⁻³, r = 0.64) and Paris (MB = -7.7 µg·m⁻³, RMSE = 13 µg·m⁻³, r = 0.78). Milano is the location with the largest MB (-14 µg·m⁻³) and RMSE (22 µg·m⁻³). The relatively low performance in Milano may be related with the inability of reproducing the strong atmospheric stability conditions of the Po Valley region, a general problem for chemical transport models. After the implementation of the national lockdowns, a decrease in $NO_2$ is simulated in all sites for both the *covid19_traffic* and

*covid19_all* scenarios. Nevertheless, the decreasing rate strongly varies from one country to the next. In Madrid and Paris, $NO_2$ concentrations drop abruptly just a few days after the beginning of the lockdown, while in Milano, Berlin and London the decreases occur at a slower pace. These results are consistent with the traffic activity reduction trends computed for these countries (Fig. 3.c). The statistics computed for the most restrictive lockdown period (23 March to 26 April) clearly reveal a general improvement of the model performance when the emission reductions are considered. As shown in Fig. 12, the

calculated MB and RMSE values for the *baseline* scenario are significantly reduced when considering the *covid19_traffic* and *covid19_all* scenarios, especially in Madrid, Paris and London where overestimations of 9 to 14 µg·m⁻³ are drastically reduced to 1 to -1.5 µg·m⁻³. In Berlin the performance of the model slightly decreases when considering the lockdown scenarios. Both the MB and RMSE of the *baseline* scenario remain lower in magnitude. This feature is attributed to a significant increase in observed $NO_2$ during the week of 7 April that neither the *baseline* nor the *covid* scenarios capture, either due to missed emission

activity changes or errors in meteorology. In terms of correlation, no significant changes are observed when comparing the *baseline* and *covid* scenarios (for all cases except Milano values stay above 0.6). The computed absolute and relative decreases of $NO_2$ urban background levels reveals that the differences between the *covid19_traffic* and *covid19_all* scenarios are generally low, i.e. the decrease in modelled $NO_2$ concentrations is mainly driven by reduction of road traffic emissions. This is consistent with the large contribution of the traffic sector to total $NO_x$ emission reductions as discussed in Sect. 4.1.

According to the modelling results, the largest decreases in urban background $NO_2$ levels occur in Madrid (-58% and -51% for *covid19_all* and *covid19_traffic*, respectively) and Milano (-56% and -54%), followed by Paris (-41% and -32%), Berlin (-30% and -23%), London (-28% and -25%) and Sweden (-11% and -10%). Among these, Paris and Berlin are the locations where non-traffic sources contribute more to total $NO_2$ reductions (around 23% in both cases).

When it comes to rural background levels, pre-lockdown statistics also indicate a good capability of MONARCH in reproducing observed values, particularly in France (MB = 1.5 µg·m⁻³, RMSE = 2.8 µg·m⁻³, r = 0.93) and Spain (MB = -0.16 µg·m⁻³, RMSE = 1.3 µg·m⁻³, r = 0.52). A persistent overestimation is observed in Germany, UK and Sweden (MB between 3.5 and 4.6 µg·m⁻³), while in Italy the system tends to underestimate (MB = -3.6 µg·m⁻³). The overestimation in Germany, UK and Sweden occurs mainly at night-time (not shown). Similar to what is observed at urban background sites, modelled and

observed concentrations between 23 March and 26 April tend to be more in agreement when considering the emission reduction scenarios. The UK and Germany are the countries were the performance improves more, with MB values going from 7.5 and 2.3 µg·m⁻³ (*baseline*) to 3.4 and 0.42 µg·m⁻³ (*covid19_traffic*) and 3 and 0.29 µg·m⁻³ (*covid19_all*). On the other hand, the



improvement is not obvious in Italy, as the model shows a negative bias during the pre-lockdown period and the lockdown scenarios constitutes an important reduction of the modelled values. However, the trend is in agreement with results in Spain,

France and Germany but with some additional underestimations. The rural background $NO_2$ concentrations in the two lockdown scenarios are substantially lower than in the *baseline* run. Nevertheless, the relative decreases are generally lower than in urban environments. France (-44% and -42% for *covid19_all* and *covid19_traffic*, respectively) and Italy (-43% and -41%) are the countries that experience the largest decreases, followed by Spain, UK and Germany (around -30% and -28% in all of them). In Sweden, relative reductions are almost equal to the ones obtained in urban background locations (-12% and -

11%). Although no robust conclusions can be extrapolated as the results are based on only one rural station, the similar reductions obtained in both environments could be related to the soft restrictions implemented in this country. When comparing the *covid19_all* and *covid19_traffic* scenarios, only around 4 to 8% of the total reduction can be attributed to non-traffic sources.

## 5    Conclusions

This paper presents a dataset of daily, sector- and country-dependent emission reduction factors that allows quantifying the impact of the COVID-19 lockdown on European primary emissions and air quality levels. The reduction factors are provided for a period that goes from 21 February, when the first European localised lockdown was implemented in the region of Lombardy (Italy), to 26 April 2020, and for the four emission sectors presumably most affected by the mobility restrictions, i.e., road transport, energy industry, manufacturing industry and aviation. Our emission reduction factors are based on a wide

range of information sources, including open access and near-real time measured activity data, proxy indicators and other available reports. We also make use of machine learning techniques trained with meteorological data to estimate reductions in electricity consumption.

We combined the computed reduction factors with the Copernicus CAMS European emission inventory using adjusted

temporal profiles in order to derive time-resolved emission reductions per country and pollutant sector. We also performed an air quality modelling study to evaluate the potential of the computed emission reductions on reproducing observed $NO_2$ concentration decreases in selected rural and urban background regions across Europe (Italy, Spain, France, Germany, UK and Sweden). The selection of countries was made considering the heterogeneous levels of lockdown restrictions and timing of implementations in each one of them. Three emission scenarios were considered: *baseline* scenario (no emission reductions

applied), *covid19_traffic* scenario (consideration of emission reductions only from road transport), and *covid19_all* scenario (consideration of emission reductions from all four sectors). Modelled results were compared against observational values reported by the EEA.

The main findings and conclusions of this work are as follows:





• During the most severe lockdown period (23 March to 26 April), estimated emission reductions at the EU-30 level were -33% for $NO_x$, -8% for NMVOC, -7% for $SO_x$ and -7% for PM2.5, with road transport being the main contributor to total reductions in all cases (85% or more) except for $SO_x$, for which reductions were mainly driven by the energy and manufacturing industry sectors.

• Italy, France and Spain are the countries that experienced the major $NO_x$ and $SO_x$ emission reductions (up to -50%

and -12%, respectively), a result that is in line with the strong lockdown restrictions implemented by their corresponding governments. On the contrary, Sweden shows reductions of only -15% ($NO_x$) and -2.5% ($SO_x$) due to implementation of national recommendations instead of a state-enforced lockdown.

• Despite showing lower reductions of road transport activity, calculated reductions of total PM2.5 in Sweden are much larger (-8%) than in Span (-4%). This is due to the variation in the contribution of the road transport sector to total

emissions from country to country. While in Sweden road transport represents around 21.3% of total PM2.5 emissions, in Spain this contribution is of just 7.9%. A similar outcome is obtained for NMVOC when comparing traffic activity and total emission reductions in Spain and France.

• According to air quality modelling results, the larger decreases of urban background $NO_2$ levels occured in Madrid (-58%) and Milano (-56%). The calculated $NO_2$ relative reductions at rural background areas are generally lower, with

France (-44%) and Italy (-43%) being the countries that experience the largest decreases.

• In both urban and rural environments, the comparison between *covid19_traffic* and *covid19_all* results, indicates that the road transport sector is on average responsible for 90% of the total $NO_2$ reductions, with the largest and lowest contributions found in Milano (97%) and Berlin (76%), respectively.

• Overall, we found the performance of the modelled $NO_2$ results to clearly improve when considering the emission

reduction scenarios. Calculated MB values for the *covid19_traffic* and *covid19_all* scenarios are significantly lower than the ones estimated for the *baseline* scenario, especially in Madrid, Paris and London where overestimations of 9 to 14 $\mu g \cdot m^{-3}$ are drastically reduced to 1 to -1.5 $\mu g \cdot m^{-3}$. On the other hand, the improvement is not so obvious at locations where the modelled results already display an important bias during the pre-lockdown period.

In this work we present and evaluate a methodology not only to calculate time-resolved emission reductions associated to the COVID-19 lockdown, but also to adapt them for air quality modelling purposes, which may be relevant for the modelling community. There are, however, some limitations associated to the current version of the reduction factors dataset. First, and most importantly, emission changes in each sector were inferred from changes observed not directly in emissions but in general activity proxies such as electricity demand or traffic indicators. The use of such general indicators may lead to disregard

changes associated to specific processes or sources. For instance, and as discussed in this work, comparisons against observed traffic counts showed that the Google movement trends are not representative of observed changes in heavy-duty vehicle's activity, and that their use may lead to a potential overestimation of the overall traffic activity reduction, especially in interurban roads, where the share of these vehicle categories is more important. In the case of energy industry, the association between





changes in electricity demand and emissions from power and heat plants neglects potential changes in the national power

mixes. As recently presented by the International Energy Agency (IEA), certain countries have shifted their electricity production towards renewables following lockdown measures due to low operating costs and priority access to the grid through regulations, among other (IEA, 2020). Omitting this aspect may be leading to an underestimation of the emission reductions for this sector, and therefore will be revised in future versions of the dataset. Finally, in the manufacturing industry sector the same reduction factors are assumed for all the industry branches. Yet, information reported by national industrial production

indexes are indicating that not all industrial sectors were affected in the same way by the lockdown restrictions. For example, Spanish pharmaceutical industries experienced no changes in their activity during March and April, while industries related to the production of mineral products showed significant decreases. Regarding this last point, it is important to note that the specificity of the computed reduction factors also depends upon the degree of sectoral disaggregation used to report the original CAMS inventory. In the case of the manufacturing industry sector, all emissions are reported under a unique category, which

hampers the consideration of industrial divisions. One last important shortcoming is related to the spatial variability of the proposed reduction factors. In its current version, the reduction factors are country-dependent and therefore do not take into account potential variations within each country. This includes, for instance, the contrast between the large cut in road traffic to and from airports on the one hand and the traffic congestion of heavy-duty vehicles at the national borders captured by the Copernicus satellite images on the other (EU, 2020). This aspect will be also relevant when extending the time series of the

dataset and including the period when governments started to soften lockdown measures. In some countries such as Spain this process was implemented heterogeneously across the different administration units.

Despite the aforementioned limitations, we believe that providing these timely emission modelling results will help with the understanding of air quality related aspects of the pandemic and also to better prepare in case of new waves or resurgences.

As a matter of fact, this dataset supports a number of studies that are on-going in particular within CAMS and under the Global Atmosphere Watch Programme of the World Meteorological Organization (WMO/GAW). Future works will focus on amending the shortcomings mentioned above, extending the number of sectors considered, in particular the residential/commercial and shipping sectors, and covering the transition period towards the post-lockdown conditions. The investigation of the calculated emission reductions obtained when combining the reduction factors with the new CAMS

emission temporal profiles (Guevara et al., *submitted*) will be also studied. New datasets and information sources will become soon available and therefore allow for an improvement of the representativeness of the current emission reductions. Moreover, the evaluation of the reduction factors in reproducing observed changes in other air pollutants such as $O_3$ or $PM_{2.5}$ will be also addressed in the future. We also expect to perform inter-comparisons of our modelled results against reductions associated to the COVID-19 lockdown derived from satellite-based observations.


**Appendix A: Quality Assurance (QA) applied to NO₂ observational dataset**


Using the information provided by GHOST (Globally Harmonised Observational Surface Treatment), we applied numerous QA screening to the NO2 dataset, in order to remove : missing measurements (flag 0), infinite values (flag 1), negative measurements (flag 2), zero measurements (flag 4), measurements associated with data quality flags given by the data provider

which have been decreed by the GHOST project architects to suggest the measurements are associated with substantial uncertainty or bias (flag 6), measurements for which no valid data remains to average in temporal window after screening by key QA flags (flag 8), measurements showing persistently recurring values (rolling 7 out of 9 data points; flag 10), concentrations greater than a scientifically feasible limit (above 5000 ppbv) (flag 12), measurements detected as distributional outliers using adjusted boxplot analysis (flag 13), measurements manually flagged as too extreme (flag 14), data with too

coarse reported measurement resolution (above 1.0 ppbv) (flag 17), data with too coarse empirically derived measurement resolution (above 1.0 ppbv) (flag 18), measurements below the reported lower limit of detection (flag 22), measurements above the reported upper limit of detection (flag 25), measurements with inappropriate primary sampling for preparing NO2 for subsequent measurement (flag 40), measurements with inappropriate sample preparation for preparing NO2 for subsequent measurement (flag 41) and measurements with erroneous measurement methodology (flag 42).

## 6    Data availability

The computed emission reduction factors per country, sector and day are provided in the supplementary material.

## 7    Author contribution

Marc Guevara conceived and coordinated the development of the European emission reduction factors. Marc Guevara and Albert Soret collected and analysed the input information required to compute the reduction factors. Hervé Petetin developed

the Machine Learning algorithm for computing business-as-usual electricity demand during the COVID-19 lockdown period. Marc Guevara and Oriol Jorba prepared the requirements for the modelling simulations. Marc Guevara, Carles Tena and Kim Serradell supervised the emission and air quality modelling simulations. Dene Bowdalo performed the evaluation of the air quality modelling results. Hugo Denier van der Gon and Jeroen Kuenen developed the CAMS-REG-AP emission inventory and have provided comments about the work. Vincent-Henri Peuch has provided comments about the work and ensured liaison

with wider activities in CAMS related to COVID-19 and air quality. Oriol Jorba and Carlos Pérez García-Pando helped conceiving the European emission reduction factors dataset and supervised the work. Marc Guevara prepared the manuscript with contributions from all co-authors.

## 8    Competing interests

The authors declare that they have no conflict of interest.



## 9   Acknowledgements

The research leading to these results has received funding from the Copernicus Atmosphere Monitoring Service (CAMS), which is implemented by the European Centre for Medium-Range Weather Forecasts (ECMWF) on behalf of the European Commission. We acknowledge support from the Ministerio de Ciencia, Innovación y Universidades (MICINN) as part of the BROWNING project RTI2018-099894-B-I00 and NUTRIENT project CGL2017-88911-R, the AXA Research Fund and the European Research Council (grant no. 773051, FRAGMENT). We also acknowledge PRACE and RES for awarding access to Marenostrum4 based in Spain at the Barcelona Supercomputing Center through the eFRAGMENT2 and AECT-2020-1-0007 projects. This project has also received funding from the European Union's Horizon 2020 research and innovation programme under the Marie Sklodowska-Curie grant agreement H2020-MSCA-COFUND-2016-754433. Carlos Pérez García-Pando also acknowledges support received through the Ramón y Cajal programme (grant RYC-2015-18690) of the MICINN.

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





**Table 1: GNFR sector classification with the definition and sources of information used to derive emission reduction factors. The countries considered for each sector are also listed.**

| Sector | Description | Sources of information | Countries included |
|---|---|---|---|
| GNFR_A | Energy industry | • Electricity demanda data: ENTSO-E (2020); FGC UES (2020)<br>• Outdoor temperature: C3S (2017)<br>• Population map: CIESIN (2016) | Austria, Belgium, Bulgaria, Croatia, Czech Republic, Estonia, France, Germany, Greece, Hungary, Ireland, Italy, Latvia, Lithuania, Netherlands, Poland, Portugal, Romania, Slovakia, Slovenia, Spain, Sweden, Switzerland, UK, Russia |
| GNFR_B | Manufacturing industry | • Electricity demanda data: ENTSO-E (2020); FGC UES (2020)<br>• Outdoor temperature: C3S (2017)<br>• Population map: CIESIN (2016)<br>• Energy balances: Eurostat (2020a) | Austria, Belgium, Bulgaria, Croatia, Czech Republic, Estonia, France, Germany, Greece, Hungary, Ireland, Italy, Latvia, Lithuania, Netherlands, Poland, Portugal, Romania, Slovakia, Slovenia, Spain, Sweden, Switzerland, UK, Russia |
| GNFR_F | Road Transport | • Movement trend reports: Google (2020) | Austria, Belgium, Bulgaria, Croatia, Republic of Cyprus, Czech Republic, Denmark, Estonia, Finland, France, Germany, Greece, Hungary, Ireland, Italy, Latvia, Lithuania, Luxembourg, Malta, Netherlands, Poland, Portugal, Romania, Slovakia, Slovenia, Spain, Sweden, Switzerland, UK, Turkey, Georgia, Bosnia and Herzegovina, Moldova, North Macedonia, Malta, Belarus |
| GNFR_H | Aviation | • Airport movement statistics: FlightRadar (2020); Eurostat (2020b) | Austria, Belgium, Bulgaria, Croatia, Republic of Cyprus, Czech Republic, Denmark, Estonia, Finland, France, Germany, Greece, Hungary, Ireland, Italy, Latvia, Lithuania, Luxembourg, Malta, Netherlands, Poland, Portugal, Romania, Slovakia, Slovenia, Spain, Sweden, Switzerland, UK, North Macedonia, Norway |



**Table 2: Absolute [µg·m⁻³] and relative changes [%] of modelled NO₂ concentrations at urban and rural background**
**stations (UB, RB) for selected countries between 23 March and 26 April. The "N" column indicates the number of stations used to compute the changes.**

| Country | Station Type | N | covid19_traffic - baseline (abs) | covid19_all - baseline (abs) | covid19_traffic - baseline (rel) | covid19_all - baseline (rel) |
|---|---|---|---|---|---|---|
| IT (Milano) | UB | 6 | -17.1 | -17.7 | -54% | -56% |
| ES (Madrid) | UB | 19 | -13.1 | -14.9 | -51% | -58% |
| FR (Paris) | UB | 16 | -8.5 | -11.0 | -32% | -41% |
| DE (Berlin) | UB | 6 | -2.9 | -3.9 | -23% | -30% |
| GB (London) | UB | 8 | -7.4 | -8.3 | -25% | -28% |
| SE (all) | UB | 8 | -0.9 | -1.1 | -10% | -11% |
| IT | RB | 69 | -2.6 | -2.7 | -41% | -43% |
| ES | RB | 58 | -0.7 | -0.8 | -28% | -31% |
| FR | RB | 23 | -2.2 | -2.3 | -42% | -44% |
| DE | RB | 74 | -1.9 | -2.0 | -26% | -28% |
| GB | RB | 14 | -3.7 | -4.0 | -28% | -30% |
| SE | RB | 1 | -0.5 | -0.6 | -11% | -12% |



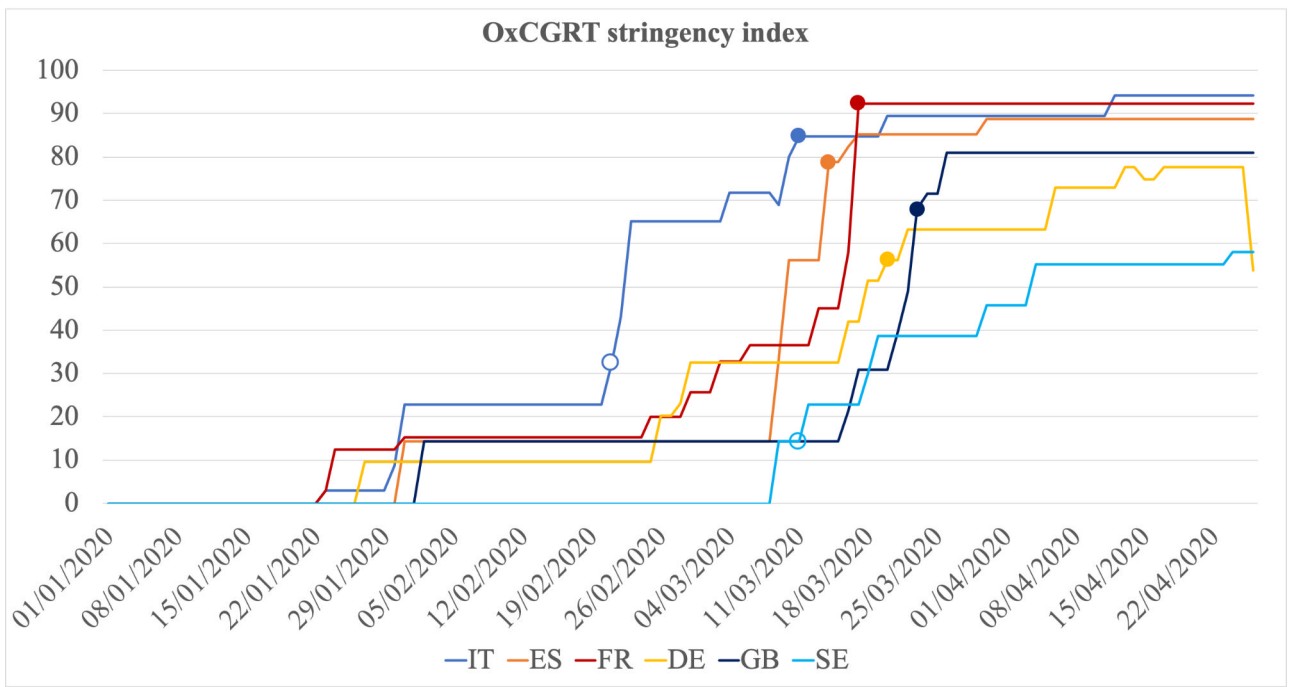

**Figure 1: Evolution of the stringency index (0 to 100) computed by the Oxford COVID-19 Government Response Tracker (OxCGRT) (Hale et al., 2020) from 1 January to 26 April 2020 for selected countries (IT, Italy; ES, Spain; FR, France; DE, Germany; GB, United Kingdom; SE, Sweden). Filled circles indicate the starting dates of national lockdowns and unfilled circles indicate the starting dates of the localised lockdown in Italy and national**

**recommendations in Sweden.**

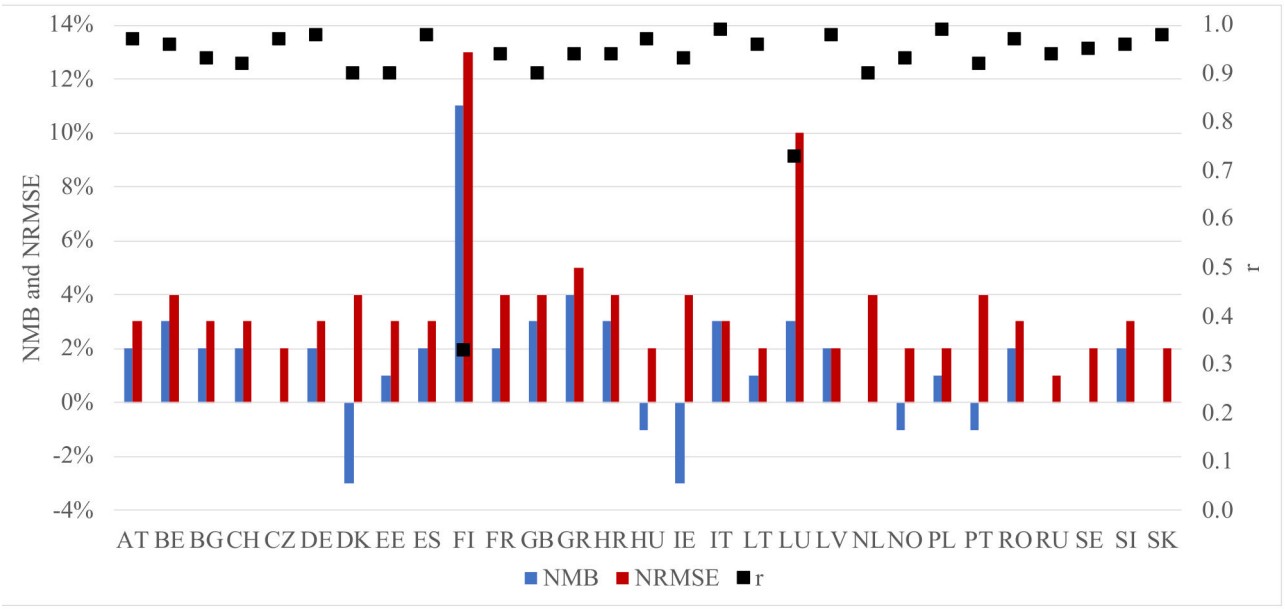

**Figure 2: Summary of the statistics (normalized mean bias, NMB; normalized root mean square error, NRMSE and correlation, r) obtained from the comparison between measured and computed electricity demand during the first two months of 2020 for selected countries.**



**Figure 3. Emission reduction factors computed for the energy (a) and manufacturing (b) industry, road transport (c) and aviation (d) for selected countries (IT, Italy; ES, Spain; FR, France; DE, Germany; GB, Great Britain; SE, Sweden) for the period 21 February to 26 April 2020. Original and COVID-19 version of the emission daily temporal factors computed for the road transport sector and used for emission modelling (e).**





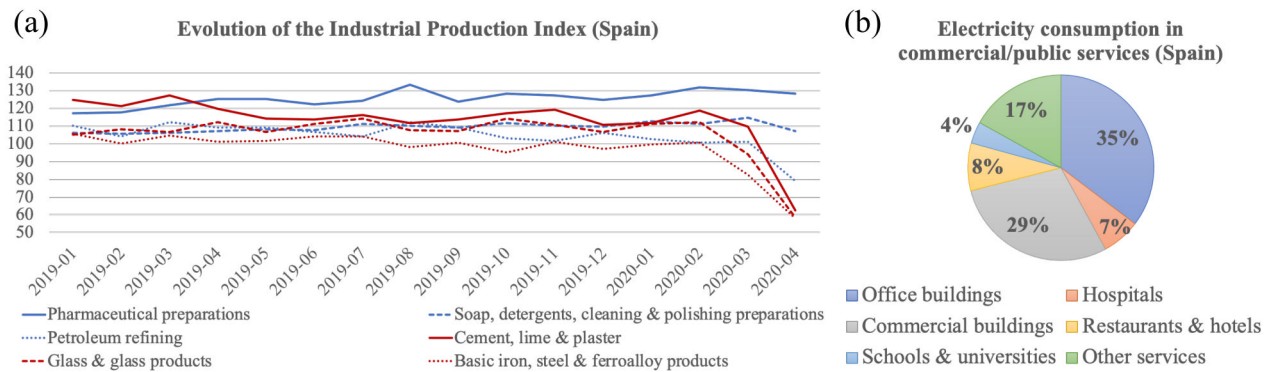

**Figure 4. Evolution of the Industrial production Index in Spain for selected manufacturing industrial branches between January 2019 and April 2020 (INE, 2020) (a). Contribution of each commercial and public service branch to total electricity consumption in Spain for 2017 (IDAE, 2018) (b).**

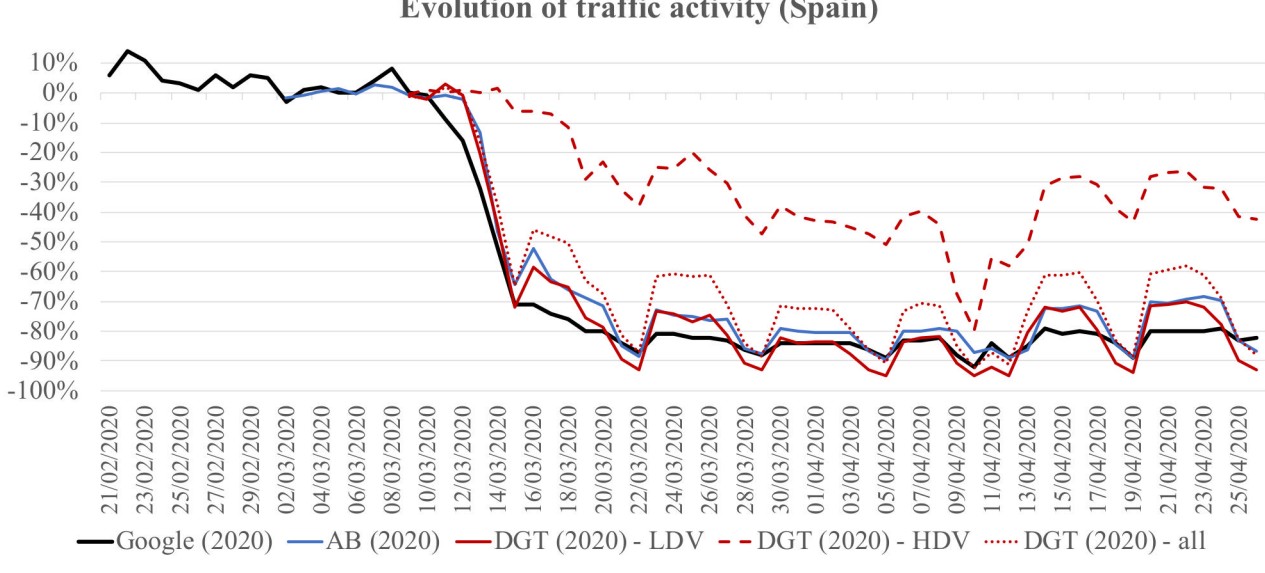

**Figure 5. Comparison of traffic movement trends for Spain derived from Google reports (Google, 2020) and measured traffic counts in the city of Barcelona (ATM, personal communication) and the main Spanish interurban roads (DGT, 2020), the latter one being also distinguished by type of vehicle (i.e. light duty vehicles, LDV; heavy duty vehicles HDV).**



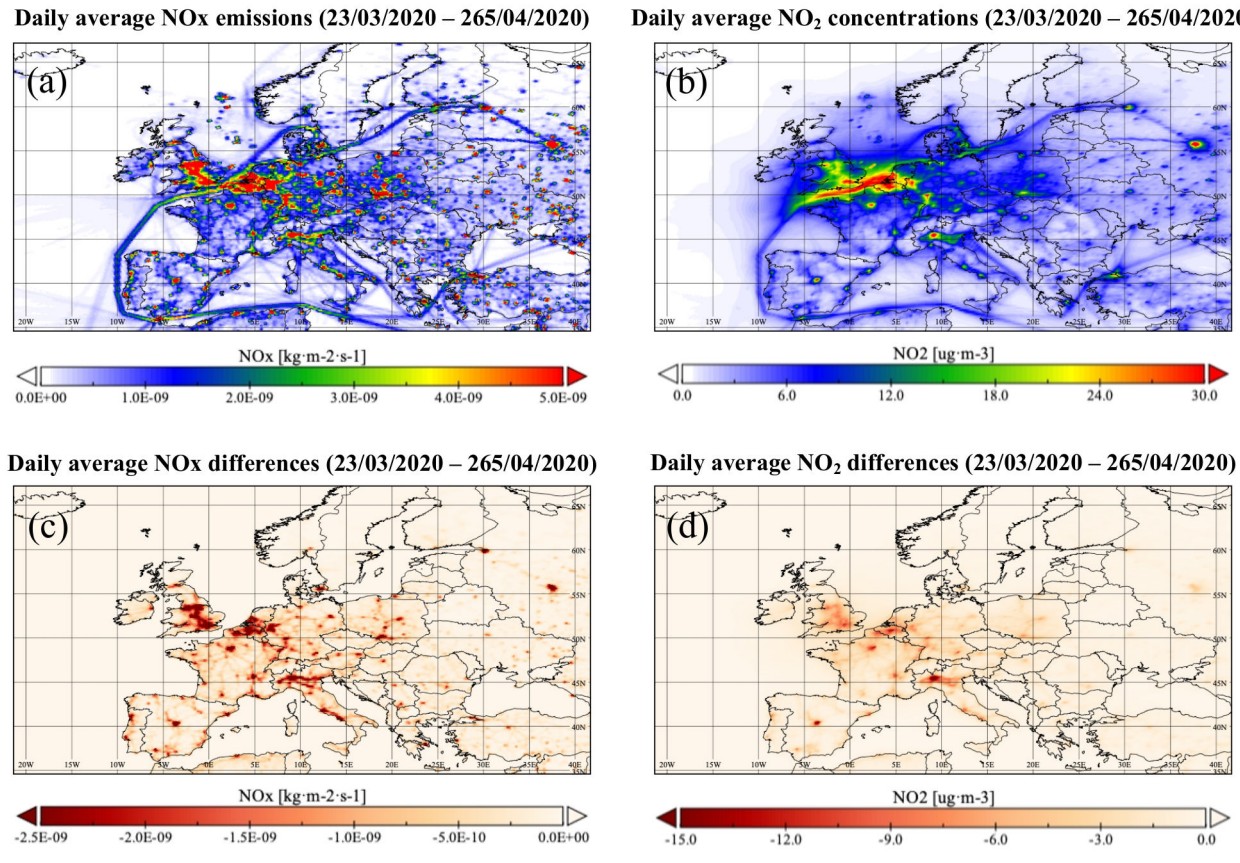

**Figure 6. Maps of the daily average NO$_x$ emissions [kg·s$^{-1}$·m$^{-2}$] (a) and NO$_2$ concentrations [µg·m$^{-3}$] (b) obtained for the** *baseline* **scenario (23 March to 26 April) and differences (c and d) when compared to the** *covid19_all* **scenario (i.e.** *covid19_all minus baseline*)**. The spatial resolution of all maps is 0.2x0.2 degrees.**







**Figure 7. Evolution in daily NOₓ (a), NMVOC (b), SOₓ (c) and PM2.5 (d) emissions [kg·day⁻¹] during the entire period of study (20 January to 26 April) for EU-30 and for each of the emission scenarios (*baseline*, *covid19_traffic* and *covid19_all*). Average (black) and 5th/95th percentiles (p05/p95) (light blue shading) relative changes [%] in gridded**

**NOₓ emissions in Italy (e) and Sweden (f) for the period 21 February to 26 April. The changes are computed considering the differences in total emissions reported by the *covid19_all* and *baseline* scenarios.**



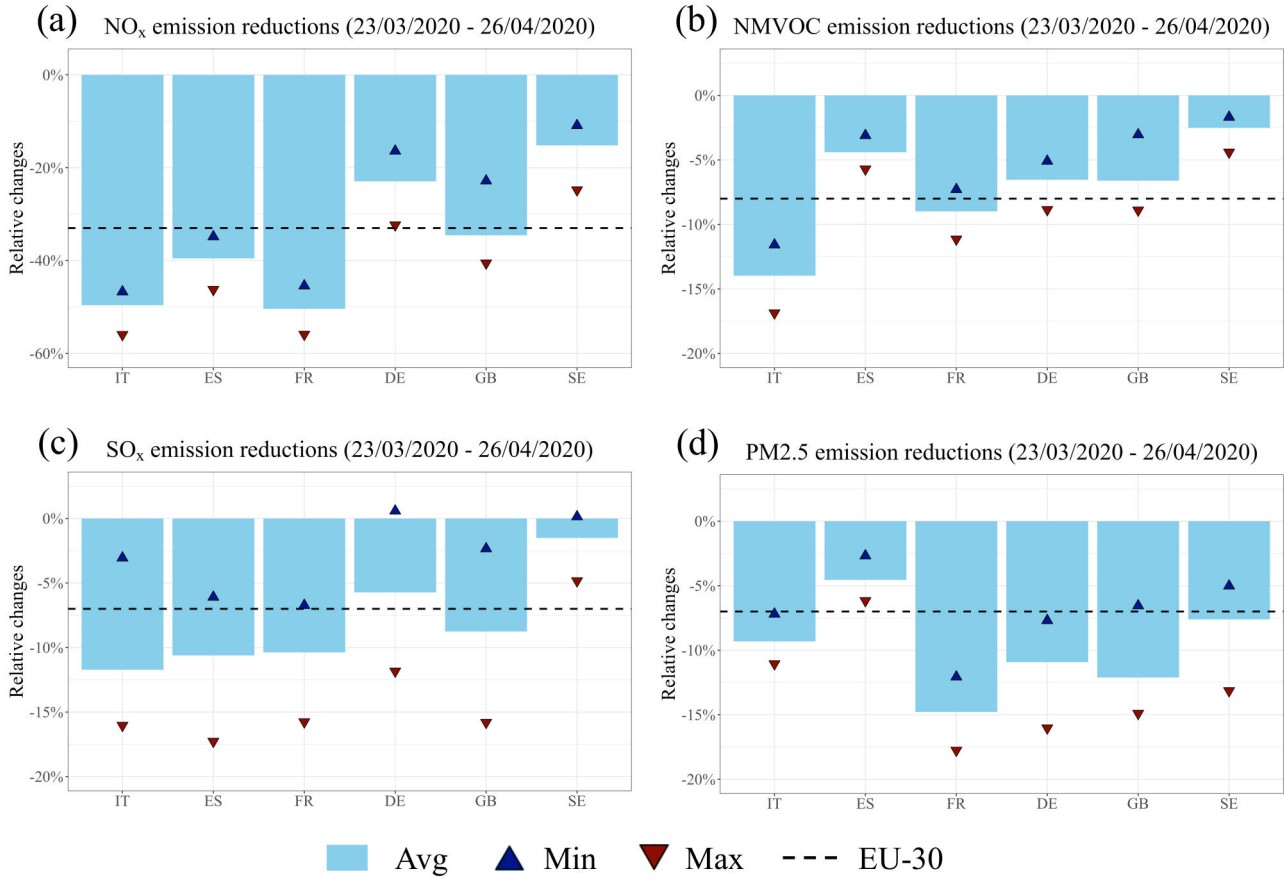

**Figure 8. Average (Avg), maximum (Max) and minimum (Min) relative changes [%] in total national NOx (a), NMVOC (b), SOx (c) and PM2.5 (d) emissions for selected countries (IT, Italy; ES, Spain; FR, France; DE, Germany; GB, United Kingdom; SE, Sweden) between 23 March and 26 April. The dashed lines indicate the relative changes at the EU-30 level. The changes are computed considering the differences in total emissions reported by the *covid19_all* and *baseline* scenarios.**



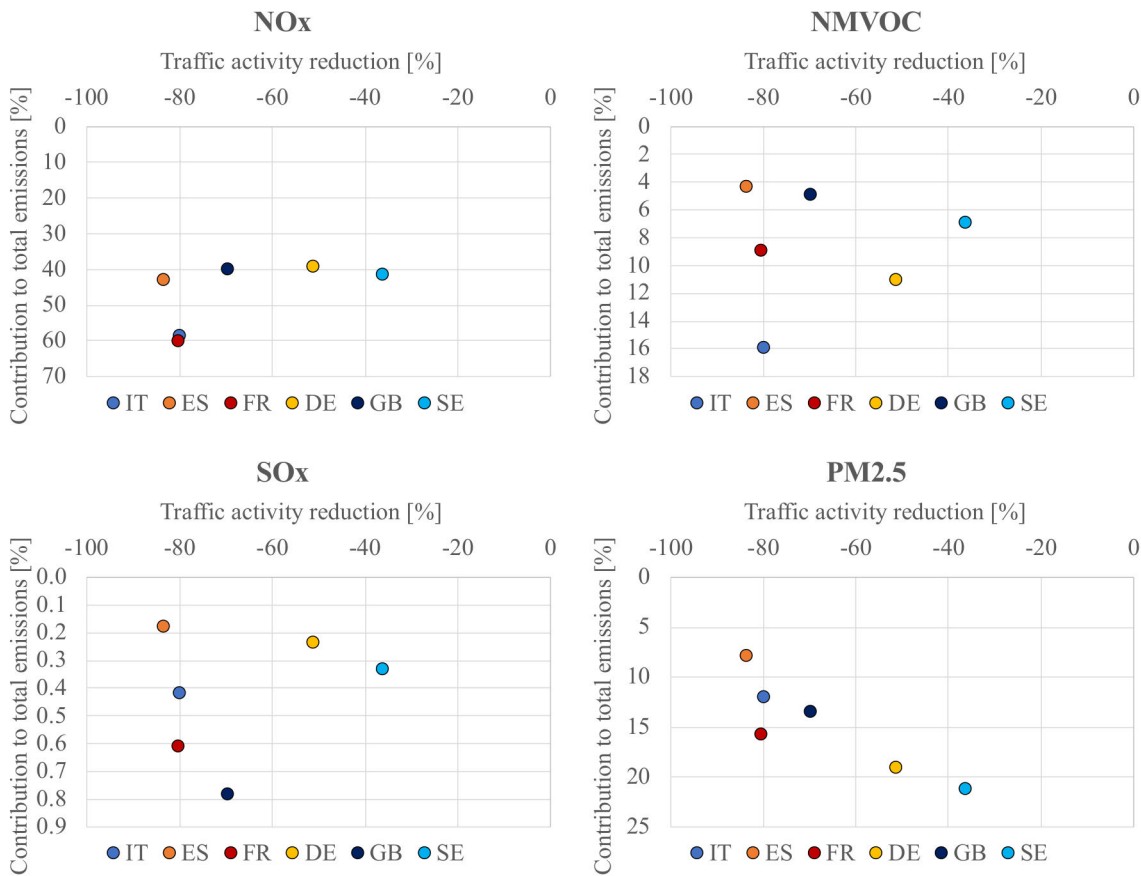

**Figure 9. Relationship between the reduction of traffic activity (23 March to 26 April) and contribution of the road transport sector to total emissions per country (IT, Italy; ES, Spain; FR, France; DE, Germany; GB, Great Britain; SE, Sweden) and pollutant (NO$_x$, NMVOC, SO$_x$, PM$_{2.5}$)**



**Figure 10. Observed (black) and modelled hourly NO₂ concentrations [μg·m⁻³] (20 January to 26 April) at selected urban background sites, including: Milan (a), Madrid (b), Paris (c), Berlin (d), London (e) and Sweden (f, all available sites). Modelled results are presented separately for each of the emission scenarios considered:** *baseline* **(in magenta),** *covid19_traffic* **(in green) and** *covid19_all* **(in purple).**

**Figure 11. Observed (black) and modelled hourly NO₂ concentrations [µg·m⁻³] (20 January to 26 April) at selected rural background sites, including: Italy (a), Spain (b), France (c), Germany (d), United Kingdom (e) and Sweden (f). Modelled results are presented separately for each of the emission scenarios considered: *baseline* (in magenta),**

***covid19_traffic* (in green) and *covid19_all* (in purple).**



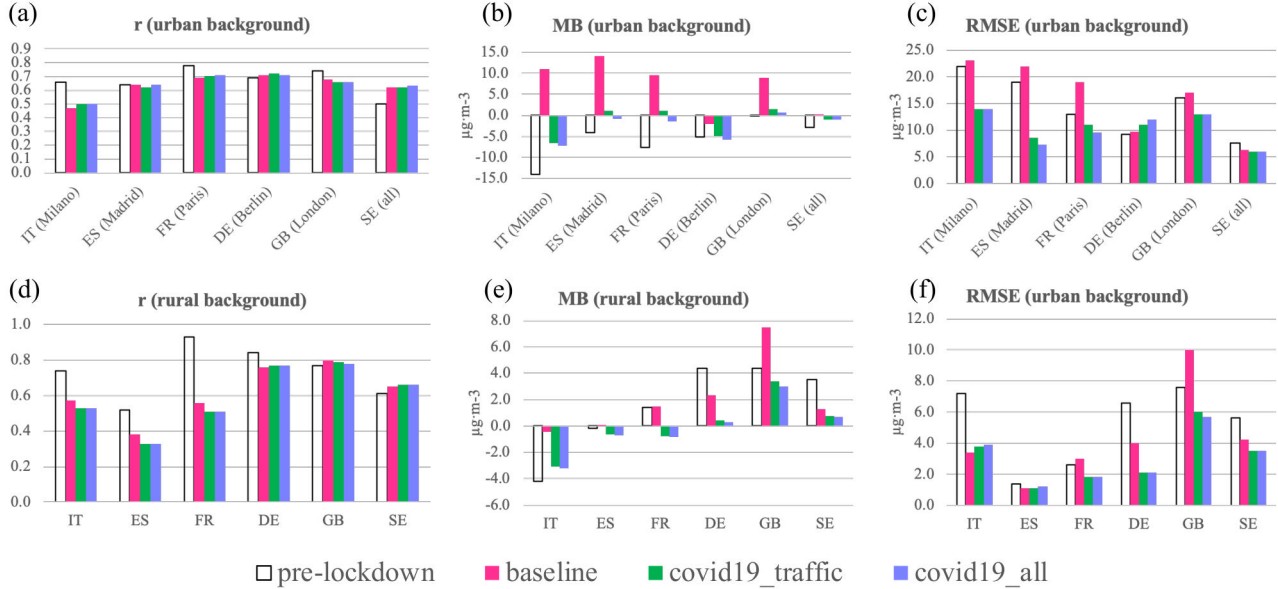

**Figure 12. Statistics calculated for NO₂ on an hourly basis for the pre-lockdown period (20 January to 20 February) and the strictest lockdown period (23 March to 26 April) at urban (a, b, c) and rural (d, e, f) background stations for selected countries (cities). Statistics calculated for most severe lockdown period are reported separately for each emission scenario (baseline, covid19_traffic and covid19_all), while for the pre-lockdown period this distinction is not made as the same emissions were used in all scenarios. The calculated statistics are mean bias (MB, μg·m⁻³), root mean square error (RMSE, μg·m⁻³) and correlation coefficient (r).**