# Peer review of "Time-resolved emission reductions for atmospheric chemistry modelling in Europe during the COVID-19 lockdowns"

_Atmospheric Chemistry and Physics, 2020_

## Referee Comment (RC1) · Anonymous Referee #1 · 17 Sep 2020

The lockdowns instituted by many governments around the world in response to the COVID-19 pandemic have had significant effects on emissions of air pollutants and resulting ambient air quality. This topic has already received a lot of attention in the scientific literature within a relatively short period of time. The manuscript by Guevara et al. provides a timely contribution to the quantification of the emission changes due to lockdown measures implemented in Europe. Traditionally, the compilation of emission inventories is a long, slow process, with reliable emission data usually becoming available after several years. Given the strong interest from modelling groups in simulating the effects of these lockdowns on air quality, there is clearly a need for a fast-track estimate of COVID-19-related changes in emissions for use in modelling studies.

[Figure]

Guevara et al. compile a set of national, sectoral emission reduction factors for European countries based on various datasets which are available now. The methodology used to derive the reduction factors is clearly described, the contingent nature of the resulting reduction factors is acknowledged and clearly described, and the reduction factors themselves are provided for the community. This aspect alone makes the paper a valuable contribution to the literature.

Guevara et al. also apply these emission reduction factors in a model simulation and compare the reductions in modelled NO2 with observed reductions in selected European cities during the lockdowns. The analysis of the model simulations is relatively superficial, but the value of the paper is clearly in the transparent calculation of the reduction factors and the provision of these factors to the community.

I only have one minor comment. The authors should indicate the year on which the CAMS-REG-AP emission inventory used in the modelling component of the study is based.

---

## Referee Comment (RC2) · Anonymous Referee #2 · 28 Sep 2020

The authors estimated the daily reductions in air pollutant emissions due to COVID-19 in Europe and evaluated the time-resolved emissions data through air quality model simulations of NO2. Activity indicators including electricity demand, heating degree day, and Google mobility reports are used in this study to represent the relative changes in emissions from different source sectors. The comparisons between simulated and observed NO2 concentrations suggest the improvement of modeling results driven by the daily emission reduction factors based on the activity indicators. This paper provides important results on the effect of COVID-19 on anthropogenic emissions and air quality, which is a hot topic at present not only in Europe but also in the other continents. Overall, I think this paper deserves publication in ACP but I still have concerns

about the uncertainties in the method and results. I suggest that the authors carefully clarify the uncertainties in the method and add a specific section in the main text to discuss the uncertainties in detail.

Major comments:

1. Energy industry. The electricity demand is estimated to have increased during COVID-19 over the Northern European countries such as Denmark, Norway, and Sweden. Why did this happen? Are these weird results relevant to the errors in the ML models designed to account for the influence of temperature fluctuations on electricity demand? Are there any temperature anomalies over North Europe during the COVID period? The authors did not explain the potential errors in the method but just assumed a null reduction of the electricity demand in Denmark, Finland, and Norway (Lines 178 to 180 in Page 6), which is not acceptable in my opinion.

2. Manufacturing industry. This study attributed 25% of the total electricity demand reduction to the reduction in manufacturing industry activity, which is rather arbitrary. What is the uncertainty involved in using such a uniform factor for the industry sector in different European countries? The authors said that the manufacturing industry sector has maintained certain activities during the COVID-19 pandemic (Line 215 in Page 7), which is not consistent with what I saw in Fig. 4a. The production of cement, iron, steel, and glass all declined significantly during April 2020 in Spain.

3. Road transport. The authors acknowledged that the emission reduction factors for the traffic sector may be overestimated because the activity levels of heavy-duty vehicles on interurban roads did not decline as much as those light-duty vehicles on urban roads. This could be the largest source of uncertainty in this study because the transport sector is the major source of NOx emissions and the heavy-duty vehicles account for a large part of transport emissions. I suggest that the authors provide more discussions on this uncertainty and try to reduce it if possible.

4. Modeling results. This study evaluated modeled NO2 concentrations with observations (Fig. 12) during the pre-lockdown and lockdown periods, respectively, which is very helpful to understand the uncertainties in the estimates of daily emissions. I suggest the authors add another figure that compares the observed and simulated NO2 decline from pre-lockdown to lockdown periods, which gives the audience more information on the accuracy of the estimated emission reduction factors.

5. Conclusions. The conclusion section is not organized well. Some paragraphs repeated the text from the method section, such as lines 509 to 517 in Page 16. Besides, the discussions on the uncertainties are not the conclusions of this study and should be written in a specific new section. Please remove the unnecessary text in the conclusions, add a new section of 'Uncertainties', and provide a condensed conclusion section.

---

## Author Comment (AC1)

We would like to thank the reviewers for their positive and constructive feedback, which helped improving the quality of the paper. The reviewers have pointed out issues that required further improvements or explanations. Below we address each specific issue and the manuscript has been updated accordingly.

**Anonymous Referee #1**

The lockdowns instituted by many governments around the world in response to the COVID-19 pandemic have had significant effects on emissions of air pollutants and resulting ambient air quality. This topic has already received a lot of attention in the scientific literature within a relatively short period of time. The manuscript by Guevara et al. provides a timely contribution to the quantification of the emission changes due to lockdown measures implemented in Europe. Traditionally, the compilation of emission inventories is a long, slow process, with reliable emission data usually becoming available after several years. Given the strong interest from modelling groups in simulating the effects of these lockdowns on air quality, there is clearly a need for a fast-track estimate of COVID-19-related changes in emissions for use in modelling studies

Guevara et al. compile a set of national, sectoral emission reduction factors for European countries based on various datasets which are available now. The methodology used to derive the reduction factors is clearly described, the contingent nature of the resulting reduction factors is acknowledged and clearly described, and the reduction factors themselves are provided for the community. This aspect alone makes the paper a valuable contribution to the literature.

Guevara et al. also apply these emission reduction factors in a model simulation and compare the reductions in modelled NO2 with observed reductions in selected European cities during the lockdowns. The analysis of the model simulations is relatively superficial, but the value of the paper is clearly in the transparent calculation of the reduction factors and the provision of these factors to the community.

I only have one minor comment. The authors should indicate the year on which the CAMS-REG-AP emission inventory used in the modelling component of the study is based.

The reference year of the CAMS-REG-AP emission inventory used is 2016, which is the most recent year available at the time of the study. We have added this information in the revised version of the manuscript as follows:

"*The base year of the CAMS-REG-APv3.1 emissions used in the three scenarios is 2016, which was the most recent year available at the time of the study*" (lines 370-371 of the revised manuscript)

Anonymous Referee #2

The authors estimated the daily reductions in air pollutant emissions due to COVID-19 in Europe and evaluated the time-resolved emissions data through air quality model simulations of NO2. Activity indicators including electricity demand, heating degree day, and Google mobility reports are used in this study to represent the relative changes in emissions from different source sectors. The comparisons between simulated and observed NO2 concentrations suggest the improvement of modeling results driven by the daily emission reduction factors based on the activity indicators. This paper provides important results on the effect of COVID-19 on anthropogenic emissions and air quality, which is a hot topic at present not only in Europe but also in the other continents. Overall, I think this paper deserves publication in ACP but I still have concerns about the uncertainties in the method and results. I suggest that the authors carefully clarify the uncertainties in the method and add a specific section in the main text to discuss the uncertainties in detail

Major comments:

1. Energy industry. The electricity demand is estimated to have increased during COVID-19 over the Northern European countries such as Denmark, Norway, and Sweden. Why did this happen? Are these weird results relevant to the errors in the ML models designed to account for the influence of temperature fluctuations on electricity demand? Are there any temperature anomalies over North Europe during the COVID period? The authors did not explain the potential errors in the method but just assumed a null reduction of the electricity demand in Denmark, Finland, and Norway (Lines 178 to 180 in Page 6), which is not acceptable in my opinion.

Regarding the case of Sweden, as mentioned in the manuscript, we hypothesize that the obtained increase in electricity demand is due to the combination of the two factors: (i) the electricity demand from public and commercial services may have remained unperturbed as there was no enforced lockdown in contrast to most other countries and (ii) the voluntary self-isolation of a fraction of the population may have increased household electricity consumption.

The reasons for assuming a null reduction of electricity demand for the other countries mentioned by the reviewer are detailed below.

In this study, we used a ML method for predicting the fluctuations of electricity demand based on temperature, assuming that temperature is a strong driver of electricity demand (for heating and air conditioning). However, temperature is obviously not the only driver of electricity demand variability, which can be influenced by various other factors (e.g. change of technology, behaviour, regulation). In addition, the gradient boosting machine models used in this study are non-parametric, meaning that they cannot extrapolate, i.e. predict electricity demand values outside the range of values used during the training phase. As a consequence, such models may perform poorly when overly strong trend and/or inter-annual variability (not directly due to temperature variability) are affecting the target variable of interest. In practice, the results obtained in this study show that this approach performs relatively well in most countries, although exceptions cannot be excluded, as shown by the case of Finland.

In Finland, the electricity demand reported by ENTSO-E in early 2020 (around late January/early February) was substantially lower than during all previous years. However, this anomaly in the power data cannot be explained by a drastic change in the temperature, as this parameter remained within the same range of values than during previous years. In such a situation, where changes in power demand cannot be related to changes in temperature, the ML cannot produce accurate predictions. We have included the following multi-panel plot (with time series and scatter plot) in the Supplement

material (Fig. S1) to illustrate the behaviour observed in the input data. Besides temperature, electricity demand in Finland is thus likely driven by other factors not included in our ML framework. Improving the predictions of electricity demand would require more complex models including heterogeneous socio-economic data, which is far beyond the scope of the present study. For this reason, we decided to discard the use of ML for this country and assume a null reduction of emissions, given that no strong lockdown was imposed to the population and no clear reduction of electricity demand was observed during the lockdown period.

[Figure]

**Figure S1. Time plots representing 7-day running averages of the electricity demand [MW] (ENTSO-E, 2020) and population-weighted outdoor temperature [ºC] (C3S, 2017) in Finland for the years 2015 until 2020 (left) and corresponding scatterplot between both variables (right). The electricity demand data represented in the time plots include ENTSO-E reported values for 2015-2020 (OBS) and the business as usual 2020 values predicted with the ML algorithm (BAU). The values of the scatterplot with the "lockdown" label (non-filled symbols) correspond to the period of the year 15/03-31/05, while the others correspond to the rest of the year. The data corresponding to the 2020 period in which negative anomalies in the electricity demand data were detected are highlighted with a dark red outline.**

A relatively similar situation was observed in Denmark. We found higher-than-usual electricity demand levels reported by ENTSO-E in late February/early March 2020 which, as in the case of Finland, could not be explained by drastic changes in temperature. At this time of the year, such relatively high power demand were already observed in 2018 but because of strong cold waves, while temperature was not particularly cold in 2020. Similar to the Finland case, we also included the following multi-panel plot in the supplementary material (Fig. S2). Thus, like in Finland, other (non-meteorological) factors are likely driving this substantial increase of electricity demand in Denmark, which explains the bias obtained from mid-February to mid-March. Without additional sources of information regarding this, we assumed again a null emission reduction.

[Figure]

**Figure S2. Similar to Fig. S1 but for Denmark. The period with strongest discrepancies between observed and predicted electricity demand before the lockdown is indicated in red.**

In Norway, although the mean bias over the entire pre-lockdown period in 2020 (2020/01/01-2020/03/15) is low (and comparable to the biases obtained in the other countries), looking at the time series shows that the bias was low at the begining of the period but started to increase in mid-February (i.e. well before the lockdown), and persisted (with some variability) during the lockdown period. The reliability of these predictions is thus lower. In addition, the increase in electricity demand obtained during the lockdown period was found to be in the same order of magnitude of the bias found before. Considering again the fact that COVID-19-related mobility restrictions were relatively soft in this country, we preferred to discard the use of ML for this country and assume a null reduction of electricity demand.

[Figure]

**Figure S3. Similar to Fig. S1 but for Norway.**

For these different countries, we do not expect that the assumed null reductions of the electricity demand will cause a significant impact on the computed emission reductions, as the majority of the electricity production in these countries comes from renewable energy sources. For instance, in the case of Norway more than 90% of the electricity production comes from hydropower (IEA, 2020). Moreover, the assumption of a null reduction for these countries is in line with the very low average changes in electricity demand reported by Le Quéré et al. (2020) (e.g. Finland, -2%, Denmark +1%).

All this information has been included in the revised version of the manuscript as follows:

*"In this study, ML models are used for predicting the fluctuations of electricity demand based on temperature (and additional time features), which assumes that temperature is a strong driver of electricity demand (for heating and air conditioning). However, temperature is obviously not the only driver of electricity demand variability, which can be influenced by various other factors (e.g. change of technology, behaviour, regulation). In addition, the gradient boosting machine models used in this study are non-parametric, meaning that they cannot extrapolate, i.e. predict electricity demand values outside the range of values used during the training phase. As a consequence, such models may perform poorly when a strong trend and/or inter-annual variability (not directly due to temperature variability) are affecting the electricity demand to predict. In practice, the results obtained in this study show that this approach performs well in most countries, although there are some exceptions. The poorest performance was obtained in Finland (r = 0.33), due to a strong negative anomaly (-12% on average) in electricity demand during January-February 2020 compared to previous years used for training. As shown in Fig. S1, the electricity demand reported by ENTSO-E for this country in early 2020 (i.e. late January/early February) was substantially lower than during all previous years (2015-2019). However, this anomaly in the power data cannot be explained by a drastic change in the temperature, as this parameter remained within the same range of values than during previous years. In such situations, where changes in power demand cannot be related to changes in temperature, the ML cannot produce accurate predictions. Compared to most other countries, a larger NRMSE and lower correlation was also found in Luxembourg. In this case, we attribute the low performance of the ML algorithm to the large data gap found in the historical data used for training. For instance, for the year 2019 the ENTSO-E dataset presents a temporal coverage lower than 50%. In addition, despite relatively good statistics in early 2020, the electricity demand computed in Denmark and Norway shows a substantial and unexpected increase during the COVID-19 lockdown (up to +12%). In the case of Denmark, we found higher-than-usual electricity demand levels reported by ENTSO-E in late February/early March 2020 which, as in the case of Finland, could not be directly explained by drastic changes in temperature (Fig. S2). At this time of the year, such relatively high power demand were already observed in 2018 but because of strong cold waves, while temperature was not particularly cold in 2020. Like in the case of Finland, unexplained changes in the electricity demand induce errors in the predictive ML algorithm. For Norway, although the mean bias on the entire test period is relatively low, a closer look to the time series indicates that this bias was low at the beginning of the period and started to increase in mid-February and persisted during the lockdown (Fig. S3). Therefore, it is unclear to which extent the increase of electricity demand during the lockdown is real or simply the persistence of the bias previously observed before the lockdown starts (as both are in the same order of magnitude). Without additional sources of information and given the relatively soft mobility restrictions imposed in Norway, we also discarded the use of ML for this country and assumed that electricity demand during the lockdown period was not significantly impacted.*

*Considering all of the above, and as a precautionary measure, we assumed a null reduction of the electricity demand in Denmark, Finland and Norway, and a fixed -16% reduction in Luxembourg starting the first day of the national lockdown implementation (15th of March), following the results reported by Le Quéré et al. (2020). Importantly, we do not expect that assuming a null reduction will cause a significant impact on the computed emission reductions, as the majority of the electricity production in these countries comes from renewable energy sources. For instance, in the case of*

*Norway more than 90% of the electricity production comes from hydropower (IEA, 2020a)."* (lines 174-210 of the revised manuscript)

*IEA: Key energy statistics. Norway. Available at: https://www.iea.org/countries/norway (last access: October 2020), 2020a.*

*Le Quéré, C., R. B. Jackson, M. W. Jones, A. J. P. Smith, S. Abernethy, R. M. Andrew, A. J. De-Gol, D. R. Willis, Y. Shan, J. G. Canadell, P. Friedlingstein, F. Creutzig and G. P. Peters: Temporary reduction in daily global CO2 emisisons during the COVID-19 forced confinement. Nature Climate Change, https://doi.org/10.1038/s41558-020-0797-x, 2020.*

2. Manufacturing industry. This study attributed 25% of the total electricity demand reduction to the reduction in manufacturing industry activity, which is rather arbitrary. What is the uncertainty involved in using such a uniform factor for the industry sector in different European countries? The authors said that the manufacturing industry sector has maintained certain activities during the COVID-19 pandemic (Line 215 in Page 7), which is not consistent with what I saw in Fig. 4a. The production of cement, iron, steel, and glass all declined significantly during April 2020 in Spain.

The attribution of 25% of the total electricity demand reduction to the reduction in manufacturing industry activity is consistent with the -27% decrease in electricity use by the manufacturing sector reported by the electricity transmission system operator of France (RTE, 2020). We have added this information in the revised version of the manuscript.

We believe the uncertainty involved in using this uniform factor is not significant compared to the uncertainty associated to the assumption of the same reduction factors for all the industry branches. We illustrated this fact in the revised version of the manuscript as follows:

*"Manufacturing industry: For this sector, the same reduction factors are assumed for all the industry branches. Yet, information reported by national industrial production indexes are indicating that not all industrial sectors were affected in the same way by the lockdown restrictions. For instance, Spanish pharmaceutical, food and paper industries experienced almost no changes in their activity during April 2020 when compared to the previous year (between 0 and -9%), while industries related to the production of petroleum and mineral products showed moderate to significant decreases in April (between -28% to -43%). For this month and country, the average reduction factor computed with the current methodology is -12.5% which, despite falling within the range of the aforementioned reductions, is not representative of the changes reported for any of the specific industrial branches. In order to overcome this limitation, specific reduction factors should be developed for each industrial branch or groups of industrial branches presenting a similar behaviour"* (lines 649-664 of the revised manuscript)

Regarding the second part of the comment, we updated Fig. 4a by adding the evolution of the Industrial production Index in Spain for the food and paper industries, which remained almost unaffected during the COVID-19 lockdowns (similar to what we were already showing for the manufacturing of pharmaceutical and cleaning products). We also clarified in the text that the industrial branches responsible of manufacturing essential goods (e.g. food, pharmaceutical preparations and other chemical products) were the ones that remained almost unaffected during the COVID-19 lockdowns.

[Figure]

**Evolution of the Industrial Production Index (Spain)**

Pharmaceutical preparations — — Food production — ■ Soap, detergents, cleaning & polishing preparations — ● Paper & paper products — Petroleum refining — Cement, lime & plaster - - - Glass & glass products ...... Basic iron, steel & ferroalloy products

3. Road transport. The authors acknowledged that the emission reduction factors for the traffic sector may be overestimated because the activity levels of heavy-duty vehicles on interurban roads did not decline as much as those light-duty vehicles on urban roads. This could be the largest source of uncertainty in this study because the transport sector is the major source of NOx emissions and the heavy-duty vehicles account for a large part of transport emissions. I suggest that the authors provide more discussions on this uncertainty and try to reduce it if possible.

We agree with the reviewer that the potential overestimation of the emissions drop from heavy duty vehicles when using the Google mobility trends may be one of the largest sources of uncertainty in this study. Therefore, an extended discussion on this topic has been added in the revised manuscript.

We used the Spanish official EMEP road transport emissions (EMEP/CEIP, 2020) which, unlike CAMS-REG-AP, are reported by vehicle category, to quantify the impact of omitting the distinction between light and heavy-duty vehicle when developing the reduction factors. We computed the evolution in daily NOx road transport emissions [t·day-1] during the entire period of study (20 January to 26 April) for Spain and for three different scenarios: (i) considering the reduction factors reported by the Google reports for Spain (Google, 2020) for all vehicle types (Google), (ii) considering the reduction factors reported by the Google reports for Spain for light duty vehicles and the ones reported by DGT (2020) for heavy duty vehicles (Google-HDV) and (iii) without considering any reduction factor (Business as usual scenario, i.e. BAU). The following figure, which has been included in the revised version of the supplementary material (Figure S4), shows the computed results:

[Figure]

**Figure S4. Evolution in daily NO$_x$ road transport emissions [t·day-1] during the entire period of study (20 January to 26 April) for Spain and for three different scenarios: (i) considering the reduction factors reported by the Google reports for Spain (Google, 2020) for all vehicle types (Google), (ii) considering the reduction factors reported by the Google reports for Spain for light duty vehicles and the reduction factors reported by DGT (2020) for heavy-duty vehicles (Google-HDV) and (iii) without considering any reduction factor (Business as usual scenario, i.e. BAU). In all three cases, the results are based on the Spanish official EMEP emissions as reported in CEIP (EMEP/CEIP, 2020)**

We compared the average emissions computed for each scenario during the strictest lockdown period (23 March to 26 April). Results indicate a -18% difference between the computed average reductions (-528.5 t when using Google trends for all vehicle categories and -434.4 t when considering specific heavy-duty vehicle trends). This difference may vary across countries due to differences in: (i) the impact of COVID-19 restriction on the activity of heavy-duty vehicles and (ii) the contribution of the heavy-duty vehicles to the overall traffic emissions.

This discussion has been introduced in the revised version of the manuscript as follows:

*"In order to quantify this uncertainty, we used the Spanish official EMEP road transport emissions (EMEP/CEIP, 2020) which, unlike CAMS-REG-AP, are reported by vehicle category, to quantify the impact of omitting the distinction between light and heavy-duty vehicle when developing the reduction factors. We compared the NOx average emission reductions obtained for the road transport sector during the strictest lockdown period (23 March to 26 April) when considering the DGT (2020) trends for heavy duty vehicles instead of the Google movement trends. Results indicate a -18% difference between the computed average reductions, i.e. -528.5 t when using Google trends for all vehicle categories and -434.4 t when considering specific heavy-duty vehicle trends (Fig. S4). This difference may vary across countries due to differences in: (i) the impact of COVID-19 restriction on the activity of heavy-duty vehicles and (ii) the contribution of the heavy-duty vehicles to the overall traffic emissions* (lines 314-321 of the revised manuscript)

We also detailed how this uncertainty could be reduced in future versions of the dataset. For that, we added the following discussion in a new subsection of the manuscript entitled "5.2 Future perspective":

*"Future works will focus on amending the shortcomings mentioned above, particularly for the case of road transport emissions and the potential overestimation of the emissions drop from heavy duty*

*vehicles when using the Google mobility trends. Measured traffic counts from other countries will be collected in order to perform an intercomparison exercise against the Google movement trends and derive a set of European adjustment factors to consider when using the original Google dataset for computing changes in emissions from heavy-duty vehicles."* (lines 683-687 of the revised manuscript)

4. Modeling results. This study evaluated modeled NO2 concentrations with observations (Fig. 12) during the pre-lockdown and lockdown periods, respectively, which is very helpful to understand the uncertainties in the estimates of daily emissions. I suggest the authors add another figure that compares the observed and simulated NO2 decline from pre-lockdown to lockdown periods, which gives the audience more information on the accuracy of the estimated emission reduction factors.

We agree with the reviewer. A new figure (Figure 13) has been added to the manuscript, which shows the comparison between observed and simulated NO$_2$ decline from pre-lockdown (20 January to 20 February) to lockdown periods (23 March to 26 April) in each region of study. Results are provided in absolute and relative terms:

[Figure]

**Figure 13. Absolute [µg·m$^{-3}$] and relative [%] observed and modelled NO$_2$ concentration declines from pre-lockdown (20 January to 20 February) to lockdown (23 March to 26 April) periods at urban (a, b) and rural (c, d) background stations for selected countries (cities).**

The discussion of the results shown in the figure were added in section 4.2 of the revised manuscript:

*"The simulated NO2 declines from pre-lockdown to lockdown periods when considering the covid19_all scenario are fairly in line with the observed ones (Fig.13.a), although a general underestimation is shown (i.e. -7.3 µg·m-3 and -6.2 µg·m-3 differences between modelled and observed declines in Italy and France, respectively). This underestimation could be related to the fact that we are currently not considering emission reductions from fuel combustion processes in commercial and institutional buildings, which were obliged to close during the lockdown period in almost all European countries."* (lines 534-543 of the revised manuscript)

*"The modelled decline of NO2 concentrations from pre-lockdown to lockdown periods presents a slight overestimation in all rural background regions except for Italy (Fig. 13.b). The largest differences occur in Germany and France, where modelled declines are 4.1 µg·m-3 and 2.8 µg·m-3 larger than the observed ones. Rural background levels can be determined by the combination of multiple emission sources and therefore it is difficult to attribute these differences to a sole reason. Nevertheless, one plausible explanation for the obtained results could be the limitation of the Google mobility trends in*

*representing the drop of emissions from heavy-duty vehicles, as discussed in Sect. 2.3"* (lines 563-569 of the revised manuscript)

5. Conclusions. The conclusion section is not organized well. Some paragraphs repeated the text from the method section, such as lines 509 to 517 in Page 16. Besides, the discussions on the uncertainties are not the conclusions of this study and should be written in a specific new section. Please remove the unnecessary text in the conclusions, add a new section of 'Uncertainties', and provide a condensed conclusion section

The conclusion section has been reorganised following the reviewer's comment. A new subsection entitled "5.1 Uncertainties" has been added to include the discussion related to the limitations and uncertainties of the present study. The original discussion has been restructured in the form of bullet points and extended considering comments #2 and #3 of the reviewer. Moreover, a new subsection called "5.2 Future perspective" has also been added, which includes the discussion related to future works. The repeated text from the method section has also been modified.